# DynaRend: Learning 3D Dynamics via Masked Future Rendering for Robotic Manipulation

**Jingyi Tian**[1]    **Le Wang**[1]*    **Sanping Zhou**[1]    **Sen Wang**[1]    **Jiayi Li**[1]    **Gang Hua**[2]

[1]National Key Laboratory of Human-Machine Hybrid Augmented Intelligence,
National Engineering Research Center for Visual Information and Applications,
Institute of Artificial Intelligence and Robotics, Xi'an Jiaotong University
[2]Amazon

## Abstract

Learning generalizable robotic manipulation policies remains a key challenge due to the scarcity of diverse real-world training data. While recent approaches have attempted to mitigate this through self-supervised representation learning, most either rely on 2D vision pretraining paradigms such as masked image modeling, which primarily focus on static semantics or scene geometry, or utilize large-scale video prediction models that emphasize 2D dynamics, thus failing to jointly learn the geometry, semantics, and dynamics required for effective manipulation. In this paper, we present **DynaRend**, a representation learning framework that learns 3D-aware and dynamics-informed triplane features via masked reconstruction and future prediction using differentiable volumetric rendering. By pretraining on multi-view RGB-D video data, DynaRend jointly captures spatial geometry, future dynamics, and task semantics in a unified triplane representation. The learned representations can be effectively transferred to downstream robotic manipulation tasks via action value map prediction. We evaluate DynaRend on two challenging benchmarks, RL-Bench and Colosseum, as well as in real-world robotic experiments, demonstrating substantial improvements in policy success rate, generalization to environmental perturbations, and real-world applicability across diverse manipulation tasks.

## 1 Introduction

Developing versatile robotic control policies capable of performing diverse tasks across varying environments has emerged as an active area of research in embodied AI [4, 3, 25, 38, 1]. Despite the promise of end-to-end approaches for generalizable robotic control, the lack of abundant, diverse and high-quality robot data remains a key bottleneck.

To address this, recent works leverage self-supervised methods to learn transferable visual representations for downstream policy learning. One line of research [30, 23, 28, 33] directly adopts 2D vision paradigms such as contrastive learning and masked image modeling. While effective at capturing high-level semantics, these methods overlook the specific needs of robotic manipulation, including 3D geometry understanding and future dynamics modeling. Another line [17, 41] focuses on predictive representations via future prediction, using large-scale video generative models to learn object and environment dynamics. However, these approaches mainly model dynamics in 2D and lack explicit awareness of the underlying 3D scene structure. A few recent efforts [26, 19] explore 3D dynamics learning with explicit representations like dynamic Gaussians, but these introduce significant structural complexity, limiting flexibility and scalability for downstream policy learning.

---

*Corresponding author.

39th Conference on Neural Information Processing Systems (NeurIPS 2025).

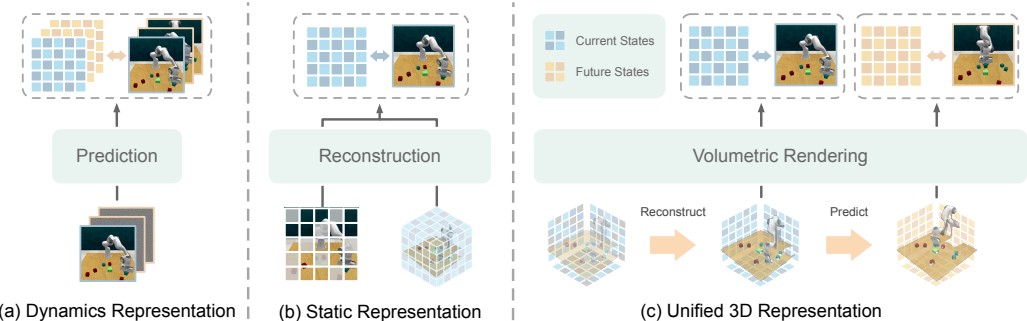

Figure 1: **Comparison of representation learning paradigms for robot learning.** (a) Learning predictive 2D representations [17] by forecasting future frames from the current observation to capture future dynamics. (b) Learning semantic or geometric features through reconstruction of static scenes using MAE [33] or 3D reconstruction [46]. (c) Our approach leverages differentiable volumetric rendering to jointly learn semantics, geometry, and dynamics in a unified 3D representation.

In this paper, we introduce DynaRend, a representation learning framework for robotic manipulation focusing on learning generalizable visual representations via 3D-aware masked future rendering. Our work leverages neural rendering to perform pretraining on multi-view RGB-D video data, enabling the model to learn representations that are grounded in both spatial geometry and future dynamics. Specifically, we begin by reconstructing a point cloud from multi-view RGB-D observations, which is then projected onto a triplane representation. A random subset of the triplane features is masked and replaced with a learnable embedding. The masked features and the language instruction are processed through a reconstructive and a predictive model, yielding two intermediate feature volumes that represent the current scene state and the predicted future counterpart, respectively. To provide supervision, we randomly select one current and one future frame, and extract their semantic features using a pretrained vision foundation model such as DINOv2 [31]. We then sample rays and apply differentiable volumetric rendering to generate RGB, semantics, and depth outputs from the predicted volumes, which are supervised by ground-truth views without annotations. After pretraining, we use the reconstruction and dynamics models to extract triplane features, which are then fine-tuned with an action decoder to predict action value maps on downstream manipulation tasks.

Compared to prior approaches, by jointly modeling spatial geometry, future dynamics, and task semantics through rendering-based supervision, DynaRend provides a unified framework for learning generalizable and scalable 3D representation tailored to manipulation tasks. Additionally, while previous methods [26, 46] incorporate neural rendering as auxiliary supervision, they typically require extensive calibrated novel views to serve as supervision, which is feasible in simulation but impractical in real-world settings with limited camera views. To address this, we leverage pretrained visual-conditioned generative models to augment target views by synthesizing novel views from existing views, reducing reliance on dense camera setups and enhancing real-world applicability.

We evaluate our method on two challenging robotic manipulation benchmarks, RLBench [21] and Colosseum [32]. Results show that our pretraining framework significantly improves manipulation success rates and exhibits strong generalization to unseen environmental perturbations such as object size, color, and lighting. We further validate our method across five real-world tasks, demonstrating effectiveness and adaptability in practical settings. Our contribution can be summarized as follows:

- We propose DynaRend, a novel representation learning framework that learns generalizable triplane features via masked future rendering for robotic manipulation.

- We conduct a systematic study of different pretraining strategies, including reconstruction and prediction objectives, masking strategy and view synthesis, on the effectiveness of downstream policy learning.

- We validate our method through extensive experiments in both simulation and the real world, showing consistent improvements over existing approaches.

## 2 Related Work

**Representation Learning for Robotics.**    In recent years, the field of computer vision has witnessed a growing research focus on self-supervised learning paradigms, where a variety of techniques [7, 9, 8, 14, 5, 31] have demonstrated remarkable effectiveness in learning transferable representations without supervision. Inspired by this, prior works have applied such techniques to robotics tasks. Notably, some approaches such as VC-1 [28], Voltron [23], and 3D-MVP [33] focus on learning discriminative visual features through masked image modeling. Other lines of work have explored the use of contrastive objectives [30] and distillation [46] for learning generalizable representations. However, existing approaches primarily focus on 2D static pretraining with limited 3D spatial awareness as well as future dynamics understanding, which are essential for manipulation tasks.

**Future Prediction for Policy Learning.**    Existing approaches have also investigated the use of future prediction as an auxiliary objective to facilitate policy learning. GR-1/GR-2 [42, 6] leverage an autoregressive transformer to generate subsequent frames and actions as a next token prediction task. SuSIE [2] and UniPi [10] employ a generative diffusion model to predict future images or video and subsequently train an inverse dynamics model conditioned on the generated goal to predict actions. VPP [17] and VidMan [41] exploit video diffusion models pretrained on large-scale datasets to capture future dynamics, which is subsequently utilized to inform action prediction. Nevertheless, existing methods primarily focus on pretraining dynamics models using 2D video data, while neglecting 3D dynamics modeling — an essential capability for robotic manipulation tasks that requires reasoning over object and environment interactions in 3D space. In addition, methods such as Imagination Policy [19] and ManiGaussian [26] have explored learning dynamics based on explicit 3D representations. Despite their ability to learn spatial and dynamic information through point clouds or 3D Gaussians, these methods often suffer from limited scalability and are difficult to directly integrate with downstream policy due to representational complexity of explicit 3D structures. In addition, the effectiveness of these methods often hinges on the availability of extensive novel-view supervision, posing significant challenges for scalability and deployment in real-world scenarios.

**Neural Rendering.**    Recent advances in 3D vision, particularly in neural rendering [29], have enabled scene representation through neural radiance fields that are supervised via volume rendering from multi-view images. In light of these developments, prior works [18, 48, 44, 39, 20] have employed differentiable neural rendering to facilitate 3D representation learning. Yet, such approaches remain confined to applications in 3D perception and autonomous driving, with limited exploration in interactive robotics scenarios. Some recent efforts have attempted to bridge this gap by applying such techniques to robot learning. For instance, SPA [47] leverages differentiable rendering to pretrain a 2D visual backbone with enhanced 3D spatial awareness; GNFactor [46] distills features from pretrained visual foundation models via volumetric rendering to learn voxel-based representations. However, these methods primarily focus on learning 3D consistency in static environments, which limits their effectiveness in robotic manipulation tasks where capturing future dynamics is essential.

## 3 Methodology

In this section, we present the proposed DynaRend in detail. We begin by formulating the problem in Sec. 3.1. In Sec. 3.2, we describe the process of feature volume extraction and construction with multi-view RGB-D image inputs. Given feature volumes, we leverage differentiable volumetric rendering to learn a reconstructive model and a predictive model for pretraining, detailed in Sec. 3.3. Finally, in Sec. 3.4, we explain how the pretrained representations and models are transferred to downstream robotic manipulation tasks. The overall pipeline is illustrated in Fig. 2.

### 3.1 Problem Definition

Language-conditioned robotic manipulation is a fundamental yet challenging task that requires agents to ground natural language instructions into executable actions based on visual observations. Among various paradigms, keyframe-based manipulation has emerged as a popular approach, where the agent is tasked with predicting the next key action state — including the end-effector pose and gripper state — given a goal-conditioned instruction and current observations. These predicted keyframes are then executed using a low-level motion planner to reach the desired target.

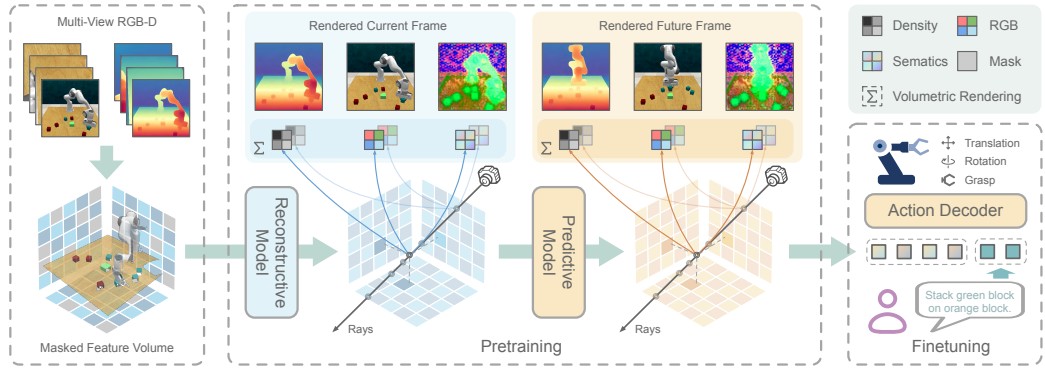

Figure 2: **DynaRend framework overview.** (a) We reconstruct the point cloud from multi-view RGB-D inputs, encode it with an MLP, and project it onto three orthogonal planes to produce triplane features. (b) We mask a subset of the triplane features and sequentially pass it through a reconstructive network and a predictive network to obtain current and future scene representations. For pretraining, both triplane volumes are rendered into RGB, depth, and semantic maps via volumetric rendering and supervised by corresponding current and future target views. (c) For finetuning, two networks serve as a triplane encoder and are trained with an action decoder on demonstration data.

One effective way to learn keyframe-based policies is through imitation learning [36, 13, 12], which allows the agent to mimic expert behaviors by learning from demonstrations. Each demonstration consists of a trajectory sequence where each element is represented as a triplet including visual observation $\mathcal{O}$, language instruction $l$, and end-effector state $\mathcal{A}$. A trajectory's keyframes can be identified via heuristic rules [36]. The learning objective for the agent is to predict the end-effector state of the nearest future keyframe, conditioned on the current observation $\mathcal{O}$ and the instruction $l$.

## 3.2 Representing 3D Scenes as Triplanes

Recent advances in 3D representation learning for robotic manipulation have explored various scene encodings, including voxel grids [36, 46], point clouds [24], and 3D Gaussians [26]. While effective in capturing geometric details, these representations are either computationally expensive or structurally complex, making them difficult to scale and generalize across tasks in manipulation-specific domains. To balance efficiency and expressiveness, we adopt a triplane representation to encode 3D scenes in a compact and learnable manner. Specifically, given the observation consisting of a set of calibrated multi-view RGB-D images $\mathcal{O} = \{I_1, I_2, \cdots, I_n\}$, we first reconstruct a scene-level point cloud using depth back-projection. The resulting point cloud is then encoded through an MLP to extract per-point features. To construct the triplane feature representation, we divide the 3D workspace into a regular voxel grid of size $H \times W \times D$, and apply axis-aligned max pooling to project the point features onto three orthogonal planes:

$$\mathbf{f}_{xy} \in \mathbb{R}^{H \times W \times C}, \quad \mathbf{f}_{xz} \in \mathbb{R}^{H \times D \times C}, \quad \mathbf{f}_{yz} \in \mathbb{R}^{W \times D \times C}, \tag{1}$$

where $C$ is the feature channel. The resulting triplane features $\mathcal{V} = \{\mathbf{f}_{xy}, \mathbf{f}_{xz}, \mathbf{f}_{yz}\}$ serve as a structured, spatially-aware encoding of the observed scene for the following reconstruction and dynamics modeling in the pretraining stage.

## 3.3 Rendering with Volumetric Representation

**Masked future prediction.** After constructing the triplane-based scene encoding, we aim to learn generalizable representations that are both 3D-aware and dynamics-informed for downstream policy learning. To this end, we formulate the pretraining task as a combination of two complementary objectives: reconstruction, which encourages understanding of scene geometry, and future prediction, which guides the model to capture future dynamics. Specifically, we first randomly mask a subset of the triplane features and replace them with a learnable mask embedding, resulting in a partially masked triplane representation denoted as $\bar{\mathcal{V}} = \{\bar{\mathbf{f}}_{xy}, \bar{\mathbf{f}}_{xz}, \bar{\mathbf{f}}_{yz}\}$. To incorporate task-specific information, we encode the language instruction using a pretrained CLIP [34] text encoder and concatenate the resulting embeddings $l$ with the triplane features, which are then fed into a reconstructive network

$\mathcal{E}_{\text{recon}}$ to reconstruct the complete 3D feature representation of the current scene:

$$\mathcal{V}_{\text{now}} = \{\mathbf{f}_{xy}^{\text{now}}, \mathbf{f}_{xz}^{\text{now}}, \mathbf{f}_{yz}^{\text{now}}\} = \mathcal{E}_{\text{recon}}(\bar{\mathcal{V}}, \mathbf{l}). \quad (2)$$

Following reconstruction, the output feature volume is further processed by a predictive network $\mathcal{E}_{\text{pred}}$ to predict the 3D representation of the nearest future keyframe:

$$\mathcal{V}_{\text{future}} = \{\mathbf{f}_{xy}^{\text{future}}, \mathbf{f}_{xz}^{\text{future}}, \mathbf{f}_{yz}^{\text{future}}\} = \mathcal{E}_{\text{pred}}(\mathcal{V}_{\text{now}}, \mathbf{l}). \quad (3)$$

Both the reconstructive and predictive networks share the same architecture and are jointly used as the triplane encoder for downstream policy learning. Each network adopts a standard Transformer architecture with four layers, enhanced with recent techniques (*i.e.*, SwiGLU [35], QK Norm [16], and RoPE [37]) to enhance stability and expressiveness.

**Volumetric rendering.** Given the reconstructed current and predicted future triplane features, we apply differentiable volumetric rendering [29] to each of them independently, using target views to supervise both the current state reconstruction and future state prediction. During training, we randomly sample a subset of pixels from a selected target view for supervision. Each pixel is associated with a camera ray $\mathbf{r}(t) = \mathbf{o} + t\mathbf{d}$, which can be defined by the camera origin $\mathbf{o}$, view direction $\mathbf{d}$ and depth $t \in [t_{\text{near}}, t_{\text{far}}]$. To render a pixel, we sample $N$ ray points $\{\mathbf{p}_i = \mathbf{o} + t_i\mathbf{d} | i = 1, \cdots, N, t_i < t_{i+1}\}$ along the ray $\mathbf{r}$. For each sampled ray point, we project its 3D coordinates onto the three axis-aligned planes (*i.e.*, the $x - y$, $x - z$, and $y - z$ planes) of the triplane features. Bilinear interpolation is then applied on each plane to query features at the projected locations. The features from the three planes are aggregated via summation to obtain a feature descriptor $\mathbf{v}_i$ for each point along the ray. These pointwise features are subsequently decoded through lightweight MLP heads to predict the attributes for each sampled point: 1) density $\sigma(\mathbf{v}_i)$: $\mathbb{R}^C \to \mathbb{R}_+$; 2) RGB values $\mathbf{c}(\mathbf{v}_i, \mathbf{d})$: $\mathbb{R}^{C+3} \to \mathbb{R}^3$; 3) sematic feature $\mathbf{s}(\mathbf{v}_i, \mathbf{d})$: $\mathbb{R}^{C+3} \to \mathbb{R}^{C'}$. Following the formulation, the final rendered outputs for each pixel are obtained by integrating the predicted attributes along the ray:

$$\hat{\mathbf{C}}(\mathbf{r}, \mathcal{V}) = \sum_{i=1}^{N} w_i \mathbf{c}(\mathbf{v}_i, \mathbf{d}), \quad \hat{\mathbf{S}}(\mathbf{r}, \mathcal{V}) = \sum_{i=1}^{N} w_i \mathbf{s}(\mathbf{v}_i, \mathbf{d}), \quad \hat{\mathbf{D}}(\mathbf{r}, \mathcal{V}) = \sum_{i=1}^{N} w_i t_i, \quad (4)$$

where $\hat{\mathbf{C}}$, $\hat{\mathbf{S}}$ and $\hat{\mathbf{D}}$ are the rendered RGB color, sematic featueres and depth respectively, $w_i = T_i(1 - \exp(\sigma(\mathbf{v}_i)\delta_i))$ is the weight for each ray point, $T_i = \exp(-\sum_{j=1}^{i-1} \sigma(\mathbf{v_j})\delta_j)$ is the accumulated transmittance, and $\delta_i = t_{i+1} - t_i$ is the distance to the previous adjacent point.

**Target view augmentation.** Existing rendering-based pretraining methods [46, 26] often rely on additional camera viewpoints as supervision, which is feasible in simulation but impractical in real-world setups due to limited camera views. To overcome this, we leverage pretrained generative models to synthesize novel views without extra cameras. Specifically, for a given target frame, we start with a set of calibrated multi-view RGB-D images. We randomly select a base view and perturb its camera pose to define a target view. The multi-view reconstructed point cloud is then warped to the target pose via projection and back-projection. We then employ See3D [27], a pretrained visual-conditioned multi-view diffusion model, to generate realistic images conditioned on the warped views, and we estimate depth maps from the synthesized views using Depth Anything v2 [45]. These generated RGB-D pairs are used as additional supervision for pretraining. In both simulation and real-world experiment setups, we rely solely on data from fixed camera viewpoints.

**Loss functions.** During pretraining, we randomly select two target views from the current frame and future frame to supervise the reconstructed and predicted triplane representations, respectively. To improve training efficiency, for each target view, we randomly sample $K$ pixels at each iteration. The rendering loss is computed as the mean squared error between the rendered outputs and the corresponding ground-truth values from the target images. To further distill high-level semantic information into triplane features, we extract semantic features from the target views using RADIOv2.5[15], an agglomerative vision foundation model, to encourage semantic consistency in the learned representations. Since our focus lies in rendering pretraining, we do not explore alternative foundation models. The rendering losses for reconstruction and future prediction are formulated as

$$\mathcal{L}_{\text{recon}} = \lambda_{\text{c}}||\mathbf{C}(\mathbf{r}) - \hat{\mathbf{C}}(\mathbf{r}, \mathcal{V}_{\text{now}})|| + \lambda_{\text{s}}||\mathbf{S}(\mathbf{r}) - \hat{\mathbf{S}}(\mathbf{r}, \mathcal{V}_{\text{now}})|| + \lambda_{\text{d}}\text{SiLog}(\mathbf{D}(\mathbf{r}), \hat{\mathbf{D}}(\mathbf{r}, \mathcal{V}_{\text{now}})),$$

$$\mathcal{L}_{\text{pred}} = \lambda_{\text{c}}||\mathbf{C}(\mathbf{r}) - \hat{\mathbf{C}}(\mathbf{r}, \mathcal{V}_{\text{future}})|| + \lambda_{\text{s}}||\mathbf{S}(\mathbf{r}) - \hat{\mathbf{S}}(\mathbf{r}, \mathcal{V}_{\text{future}})|| + \lambda_{\text{d}}\text{SiLog}(\mathbf{D}(\mathbf{r}), \hat{\mathbf{D}}(\mathbf{r}, \mathcal{V}_{\text{future}})),$$

$$(5)$$

where $\mathbf{C}(\mathbf{r})$, $\mathbf{S}(\mathbf{r})$ and $\mathbf{D}(\mathbf{r})$ denote the ground-truth RGB values, sematic features and depth respectively, SiLog is scale-invariant log loss [11] to optimize depth, and $\lambda_c$, $\lambda_s$ and $\lambda_d$ are weights to balance different losses. The overall objective for pretraining is a weighted combination of two loss terms for reconstruction and future prediction respectively:

$$\mathcal{L}_{\text{pretrain}} = \lambda_{\text{recon}}\mathcal{L}_{\text{recon}} + \lambda_{\text{pred}}\mathcal{L}_{\text{pred}}, \tag{6}$$

where $\lambda_{\text{recon}}$ and $\lambda_{\text{pred}}$ are loss weights.

### 3.4 Predicting Actions for Downstream Tasks

To adapt the pretrained triplane encoder to downstream robotic manipulation tasks, we extend it with an action decoder and fine-tune the entire model using expert demonstrations. Following RVT [13], we formulate action prediction as multi-view action value map prediction. Specifically, given current observations, inputs are passed sequentially through the reconstructive network $\mathcal{E}_{\text{recon}}$ and the predictive network $\mathcal{E}_{\text{pred}}$ without masking to extract a spatially and dynamically informed triplane representation. The objective is to predict the next keyframe action $\mathcal{A} = \{\mathbf{a}_{\text{pose}}, \mathbf{a}_{\text{gripper}}\}$, where $\mathbf{a}_{\text{pose}} = \{\mathbf{a}_{\text{trans}}, \mathbf{a}_{\text{rot}}\} \in \text{SE}(3)$ is the end-effector pose, and $\mathbf{a}_{\text{gripper}} \in \{0, 1\}$ is the gripper state.

For translation component $\mathbf{a}_{\text{trans}}$, we process the extracted triplane features through a convolution layer and an upsampling layer to produce action heatmaps over the three orthogonal planes. The ground-truth action translation is projected onto the three orthogonal planes to generate corresponding target heatmaps, which are used to supervise the predicted action heatmaps via cross entropy loss. For rotation component $\mathbf{a}_{\text{rot}}$ and gripper state $\mathbf{a}_{\text{gripper}}$, we query the triplane features at the predicted action translation position by interpolating the three planes and aggregating the resulting features by summation. The resulting feature is then passed through a lightweight MLP to predict discretized rotation Euler angles and the binary gripper open/close state, both supervised using cross entropy loss with respect to the ground-truth labels. The final fine-tuning loss is formulated as

$$\mathcal{L}_{\text{finetune}} = \lambda_{\text{trans}}\text{CE}(\mathbf{a}_{\text{trans}}, \hat{\mathbf{a}}_{\text{trans}}) + \lambda_{\text{rot}}\text{CE}(\mathbf{a}_{\text{rot}}, \hat{\mathbf{a}}_{\text{rot}}) + \lambda_{\text{gripper}}\text{CE}(\mathbf{a}_{\text{gripper}}, \hat{\mathbf{a}}_{\text{gripper}}) \tag{7}$$

where $\hat{\mathbf{a}}_{\text{trans}}$, $\hat{\mathbf{a}}_{\text{rot}}$ and $\hat{\mathbf{a}}_{\text{gripper}}$ are the predicted action translation, rotation and gripper state respectively. During inference, the predicted orthogonal plane-wise heatmaps are first broadcast and summed across axes to form a 3D heatmap over the workspace. The spatial location with the highest activation is selected as the predicted position of the end-effector for the next keyframe. This position is then used to query the triplane representation for subsequent rotation and gripper state prediction, following the same decoding procedure as during training.

## 4 Experiments

### 4.1 Simulation Experiments

**Environmental setup.**    We conduct simulation experiments on two challenging robotic manipulation benchmarks: RLBench [21] and Colosseum [32]. All experiments utilize a 7-DoF Franka Emika Panda robot arm equipped with a parallel gripper, mounted on a fixed tabletop setup. The input observation at each time step consists of four calibrated RGB-D images captured from the front, left shoulder, right shoulder, and wrist viewpoints of the robot, following prior works [36, 13]. We collect 100 expert demonstrations per task for training on both benchmarks.

For RLBench, we consider two widely adopted evaluation settings: a 18-task subset from recent works [36, 13, 12, 33], and a 71-task setting covering all executable tasks in the suite. For the latter, we follow SPA [47] to divide the tasks into two groups for evaluation. For both settings, each task is evaluated over 25 rollout episodes, and we report the average task success rate. Colosseum [32] is a benchmark for evaluating the generalization capabilities of manipulation policies under 12 types of environmental perturbations across 20 tasks, including changes in object color, texture, size, and lighting. For training, we use the 100 expert demonstrations collected in the unperturbed environment for each task. During test time, for each task, we separately apply each of the 12 perturbation types, and rollout 25 episodes per perturbation. We report the average success rate across each perturbation category to assess the robustness of the policy to different types of environmental changes.

**Comparisons and baselines.**    We compare the proposed method against various baselines across both benchmarks. For the RLBench benchmark, we compare our method against both different

| | Avg. S.R.↑ | Avg. Rank↓ | Inf. Speed↑ | Push Buttons | Slide Block | Sweep to Dustpan | Meat off Grill | Turn Tap | Put in Drawer | Close Jar |
|---|---|---|---|---|---|---|---|---|---|---|
| C2F-ARM-BC [22] | 20.1 | 6.4 | - | 72.0 | 16.0 | 0.0 | 20.0 | 68.0 | 4.0 | 24.0 |
| PerAct [36] | 49.4 | 5.1 | 4.9 | $92.8_{\pm3.0}$ | $74.0_{\pm13.0}$ | $52.0_{\pm0.0}$ | $70.4_{\pm2.0}$ | $88.0_{\pm4.4}$ | $51.2_{\pm4.7}$ | $55.2_{\pm4.7}$ |
| RVT [13] | 62.9 | 4.3 | 11.6 | $100.0_{\pm0.0}$ | $81.6_{\pm5.4}$ | $72.0_{\pm0.0}$ | $88.0_{\pm2.5}$ | $93.6_{\pm4.1}$ | $88.0_{\pm5.7}$ | $52.0_{\pm2.5}$ |
| 3D-MVP [33] | 67.5 | 3.2 | 11.6 | 100.0 | 48.0 | 80.0 | 96.0 | 96.0 | 100.0 | 76.0 |
| 3D Diffuser Actor [24] | 81.3 | 2.2 | 1.4 | $98.4_{\pm2.0}$ | $97.6_{\pm3.2}$ | $84.0_{\pm4.4}$ | $96.8_{\pm1.6}$ | $99.2_{\pm1.6}$ | $96.0_{\pm3.6}$ | $96.0_{\pm2.5}$ |
| RVT-2 [12] | 81.4 | 2.2 | 20.6 | $100.0_{\pm0.0}$ | $92.0_{\pm2.8}$ | $100.0_{\pm0.0}$ | $99.0_{\pm1.7}$ | $99.0_{\pm1.7}$ | $96.0_{\pm2.0}$ | $100.0_{\pm0.0}$ |
| **DynaRend** | **83.2** | **1.5** | 19.6 | $100.0_{\pm0.0}$ | $100.0_{\pm0.0}$ | $93.6_{\pm5.4}$ | $100.0_{\pm0.0}$ | $100.0_{\pm0.0}$ | $99.2_{\pm1.8}$ | $91.2_{\pm4.4}$ |

| | Drag Stick | Put in Safe | Place Wine | Screw Bulb | Open Drawer | Stack Blocks | Stack Cups | Put in Cupboard | Insert Peg | Sort Shape | Place Cups |
|---|---|---|---|---|---|---|---|---|---|---|---|
| C2F-ARM-BC [22] | 24.0 | 12.0 | 8.0 | 8.0 | 20.0 | 0.0 | 0.0 | 0.0 | 4.0 | 8.0 | 0.0 |
| PerAct [36] | $89.6_{\pm4.1}$ | $84.0_{\pm3.6}$ | $44.8_{\pm7.8}$ | $17.6_{\pm2.0}$ | $88.0_{\pm5.7}$ | $26.4_{\pm3.2}$ | $2.4_{\pm2.0}$ | $28.0_{\pm4.4}$ | $5.6_{\pm4.1}$ | $16.8_{\pm4.7}$ | $2.4_{\pm3.2}$ |
| RVT [13] | $99.2_{\pm1.6}$ | $91.2_{\pm3.0}$ | $91.0_{\pm5.2}$ | $48.0_{\pm5.7}$ | $71.2_{\pm6.9}$ | $28.8_{\pm3.9}$ | $26.4_{\pm8.2}$ | $49.6_{\pm3.2}$ | $11.2_{\pm3.0}$ | $36.0_{\pm2.5}$ | $4.0_{\pm2.5}$ |
| 3D-MVP [33] | 100.0 | 92.0 | 100.0 | 60.0 | 84.0 | 40.0 | 36.0 | 60.0 | 20.0 | 28.0 | 4.0 |
| 3D Diffuser Actor [24] | $100.0_{\pm0.0}$ | $97.6_{\pm2.0}$ | $93.6_{\pm4.8}$ | $82.4_{\pm2.0}$ | $89.6_{\pm4.1}$ | $68.3_{\pm3.3}$ | $47.2_{\pm8.5}$ | $85.6_{\pm4.1}$ | $65.6_{\pm4.1}$ | $44.0_{\pm4.4}$ | $24.0_{\pm7.6}$ |
| RVT-2 [12] | $99.0_{\pm1.7}$ | $96.0_{\pm2.8}$ | $95.0_{\pm3.3}$ | $88.0_{\pm4.9}$ | $74.0_{\pm11.8}$ | $80.0_{\pm2.8}$ | $69.0_{\pm5.9}$ | $66.0_{\pm4.5}$ | $40.0_{\pm0.0}$ | $35.0_{\pm7.1}$ | $38.0_{\pm4.5}$ |
| **DynaRend** | $100.0_{\pm0.0}$ | $96.0_{\pm4.0}$ | $98.4_{\pm2.2}$ | $88.0_{\pm6.3}$ | $89.6_{\pm2.2}$ | $71.2_{\pm3.3}$ | $82.4_{\pm3.6}$ | $87.2_{\pm1.8}$ | $31.2_{\pm3.3}$ | $44.8_{\pm3.3}$ | $25.6_{\pm5.4}$ |

Table 1: **Evaluation results on 18 RLBench tasks.** Each task is evaluated with 25 rollouts under 5 different seeds. We report the average success rate and standard deviation for all tasks.

policy architectures and pretraining strategies. In the 18-task setting, we primarily evaluate against previous state-of-the-art policy models. On the 71-task setting, we mainly adopt the strong RVT-2 [12] architecture for all methods and replace its Transformer backbone with various pretrained visual models to compare different pretraining strategies, including 2D pretraining methods such as MVP [43] and VC-1 [28], and 3D pretraining methods such as SPA [47]. For the Colosseum benchmark, we compare against state-of-the-art baselines that evaluate policy generalization under different domain shifts. The comparison includes three pretraining-based approaches including MVP [43], VC-1 [28] and 3D-MVP [33], as well as RVT [13] architecture training from scratch.

**Implementation details.** During both pretraining and fine-tuning stages, we apply SE(3) augmentations to the input point clouds, camera poses, and action labels. Specifically, we perform random translations along the x, y, and z axes by up to 0.125 m, and random rotations around the z-axis by up to 45 degrees. The triplane grid is constructed with a resolution of $16 \times 16 \times 16$. For pretraining, we train the model for ~60k steps, while for fine-tuning on downstream tasks, we train for an additional ~30k steps. In both stages, we use a batch size of 256 and set the initial learning rate to $1 \times 10^{-4}$ with cosine decay schedule. Training is conducted using 8 NVIDIA RTX 3090 GPUs.

**Results on RLBench.** We report the average success rates across the 18 RLBench tasks in Tab. 1. Our method significantly outperforms existing state-of-the-art approaches, including RVT-2 [12] and 3D Diffuser Actor [24]. Notably, compared to the baseline RVT [13] model, DynaRend achieves an average success rate improvement of 32.3%. Even relative to RVT-2, a two-stage variant of RVT that incorporates additional refinement, our method still shows noticeable gains. In addition to performance gains, DynaRend also exhibits superior efficiency. Our model achieves the best trade-off between success rate and inference speed when compared to other baseline methods, demonstrating strong manipulation performance without sacrificing computational efficiency.

We further evaluate the scalability of our approach on the larger 71-task RLBench setting, comparing DynaRend against RVT-2 models with various pretraining methods. The results, summarized in Tab. 2, demonstrate that DynaRend consistently outperforms existing approaches, achieving an average improvement of 8.1% over two-stage baselines. These baselines include both 2D pretraining methods (e.g., MVP [43], VC-1 [28]) and 3D pretraining methods (e.g., SPA [47]), all implemented within the two-stage RVT-2 [12] model. Additionally, DynaRend achieves a 25.2% improvement over

| Method | Group 1 (35) | Group 2 (36) | Avg. S.R. |
|---|---|---|---|
| *two-stage* | | | |
| MoCov3 [9] | 73.7 | 54.2 | 63.9 |
| MAE [14] | 78.3 | 57.7 | 68.0 |
| DINOv2 [31] | 78.2 | 56.1 | 67.1 |
| CLIP [34] | 76.8 | 55.7 | 66.2 |
| MVP [43] | 76.2 | 56.3 | 66.2 |
| VC-1 [28] | 80.1 | 55.7 | 67.9 |
| SPA [47] | 80.5 | 61.2 | 70.8 |
| *single-stage* | | | |
| RVT [13] | 71.9 | 50.4 | 61.1 |
| **DynaRend** | **81.4** | **71.8** | **76.6** |

Table 2: **Results on 71 RLBench tasks.**

the single-stage RVT baseline [13]. Moreover, unlike prior methods that rely on large-scale external pretraining datasets, our method is pretrained solely on task-relevant multi-view RGB-D data without additional external supervision. This highlights the efficiency and task-adaptiveness of DynaRend, making it practical for scalable deployment in real-world setups.

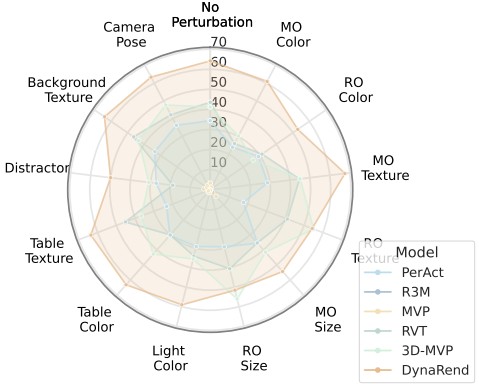

Figure 4: **Results on Colosseum.**

| Ablation | Avg. S.R.(%) | Δ |
|---|---|---|
| **DynaRend** | **83.2** | - |
| w/o. pretraining | 76.7 | -6.5 |
| w/o. reconstruction | 80.7 | -2.5 |
| w/o. future prediction | 78.9 | -4.3 |
| w/o. RGB loss | 78.2 | -5.0 |
| w/o. sematic loss | 80.4 | -2.8 |
| w/o. depth loss | 82.0 | -1.2 |
| w/o. novel view augmentation | 79.8 | -3.4 |

Table 3: **Ablations.** Average success rates are reported over RLBench 18 tasks to evaluate the impact of different design choices.

**Results on COLOSSEUM.** We present the results on the Colosseum benchmark in Fig. 4. Compared to existing 2D pretraining methods, such as MVP [43] and R3M [30], as well as 3D pretraining approaches like 3D-MVP [33], DynaRend achieves consistently higher success rates across most categories of environmental perturbations. Additionally, when compared to the RVT baseline trained from scratch, DynaRend demonstrates significantly greater robustness to various types of environmental variations. In particular, our method shows notable improvements in scenarios involving object and environment texture variations. We attribute these gains to the robust spatial and physical priors captured during the 3D-aware masked future rendering pretraining, which enables the policy to better generalize to unseen domain shifts without requiring domain randomization during training.

## 4.2 Ablation Study & Analysis

**Impact of reconstruction and prediction.** We study the effectiveness of pretraining and the contribution of each individual objective, namely reconstruction and future prediction, by selectively enabling them during the pretraining stage. As shown in Tab. 3, we report the average success rates across the 18 RLBench tasks under four configurations: (1) training the policy model from scratch without any pretraining; (2) pretraining with only the reconstruction objective; (3) pretraining with only the future prediction objective; and (4) pretraining with both objectives combined. The results indicate that pretraining consistently improves downstream policy performance, showcasing the effectiveness of the proposed pretraining strategy. Among the two objectives, future prediction yields greater gains than reconstruction alone. Importantly, combining both objectives leads to the best performance, highlighting their complementary roles: reconstruction helps the model capture spatial geometry, while future prediction encourages learning of future dynamics critical for manipulation.

**Impact of mask ratio.** Additionally, we perform an ablation study on the effect of the masking ratio applied to the triplane features in Fig. 3. Removing masking entirely or applying an excessively high mask ratio both lead to degraded performance. Introducing a moderate level of masking improves generalization by preventing overfitting to specific camera views. In contrast, a high mask ratio hampers the model's ability to reconstruct meaningful and coherent 3D representations during pretraining, leading to a larger gap between the pretraining and fine-tuning stages.

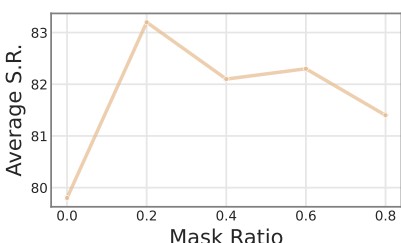

Figure 3: **Ablation on mask ratio.**

**Impact of different rendering objectives.** In Tab. 3, we compare the impact of different rendering objectives used during pretraining on downstream task performance. We evaluate the contribution of each loss term, including RGB reconstruction, semantic alignment, and depth supervision, by selectively enabling them and measuring the resulting average success rate across the 18 RLBench tasks. Our results indicate that both the RGB loss and semantic loss are critical for learning effective 3D representations, directly influencing the quality of the learned features, which in turn affects the performance of downstream policy. The depth supervision also provides moderate improvements,

though its contribution is less significant. We hypothesize that the triplane representation is derived from depth-projected point clouds, and thus already encodes explicit 3D information to some extent.

**Impact of target view augmentation.**    We further conduct an ablation on the target view augmentation strategy, as shown in Tab. 3. Incorporating synthetic views during pretraining helps mitigate overfitting to the limited camera viewpoints and encourages the model to learn more view-invariant and robust 3D representations that better capture spatial geometry. This leads to a noticeable improvement in downstream success rates, confirming the benefit of view diversity in supervision.

## 4.3   Real-world Experiments

**Environmental setup.**    We evaluate our method on five real-world robotic manipulation tasks and compare it against prior state-of-the-art approach. All experiments are conducted using a Franka Research 3 robot arm equipped with a parallel gripper, mounted on a fixed table-top setup. For visual input, we employ two calibrated Orbbec Femto Bolt RGB-D cameras, mounted to the left and right of the robot. For each task, we collect 30 expert demonstrations, with spatial configurations of objects randomized across episodes. Some tasks include multiple variants involving different target objects and instructions. Additionally, during testing, we introduce distractor objects into the

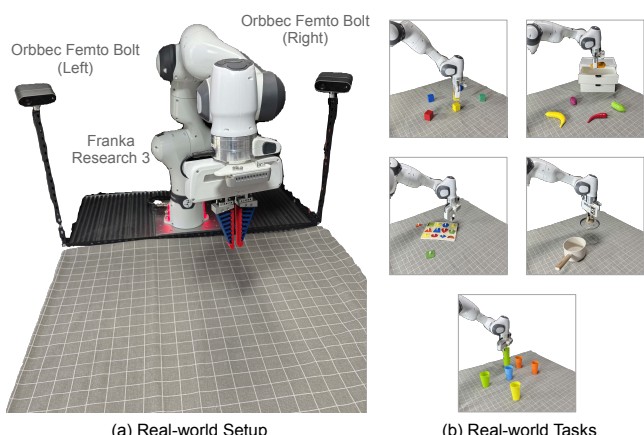

Figure 5: **Real-world setup and task examples.** We evaluate on five manipulation tasks: `Put Item in Drawer`, `Close Pot`, `Stack Blocks`, `Sort Shape`, `Stack Cups`.

scenes to further evaluate the robustness of the learned policy under challenging and diverse conditions. Detailed descriptions of each task are provided in the appendix. We pretrain our model on the collected real-world dataset for 30k steps with augmented views and fine-tune it for an additional 10k steps. The training hyperparameters are kept consistent with the simulation experiments.

**Quantitative results.**    In Tab. 4, we report the average success rates across the five real-world tasks. The results show that our method consistently outperforms prior method. Notably, on tasks involving distractor objects, RVT-2 struggles to distinguish between different unseen items, leading to frequent failure cases. In contrast, our method maintains robust performance, benefiting from the pretrained spatially grounded and semantically coherent representations. Additionally, in long-horizon tasks such as `Stack Cups`, DynaRend achieves sizable

|  | Put Item in Drawer | Stack Blocks | Sort Shapes | Close Pot | Stack Cups | Avg. S.R. |
|---|---|---|---|---|---|---|
| 3DA [24] | 40 | 50 | 30 | 45 | 20 | 37 |
| RVT [13] | 25 | 40 | 5 | 55 | 5 | 26 |
| RVT-2 [12] | 45 | **60** | 10 | 60 | 15 | 37 |
| **DynaRend** | **65** | **60** | **35** | **85** | **40** | **57** |

|  | Put Item in Drawer* | Stack Blocks* | Sort Shapes* | Close Pot* | Stack Cups* | Avg. S.R. |
|---|---|---|---|---|---|---|
| 3DA [24] | 20 | 15 | 25 | 40 | 0 | 20 |
| RVT [13] | 5 | 15 | 0 | 25 | 5 | 10 |
| RVT-2 [12] | 15 | 10 | 10 | 45 | 0 | 16 |
| **DynaRend** | **55** | **55** | **25** | **65** | **25** | **45** |

Table 4: **Real-world performance.** We report the average success rates over 20 rollouts for each task. * indicates tasks where additional distractors were introduced during testing.

performance gains, which we attribute to the joint learning of 3D geometry and future dynamics in pretraining stage, facilitating effective reasoning and manipulation in physically complex scenarios.

# 5 Conclusion and Discussion

In this work, we proposed DynaRend, a representation learning framework that unifies spatial geometry, future dynamics, and task semantics through masked reconstruction and future prediction with differentiable volumetric rendering. DynaRend learns transferable 3D-aware and dynamics-informed triplane representations from multi-view RGB-D data, which can be effectively adapted to downstream manipulation tasks via action value map prediction. Our extensive experiments on RLBench, Colosseum, and real-world setups demonstrate significant improvements in policy performance and robustness under environmental variations. These results highlight the potential of rendering-based future prediction for scalable and generalizable robot learning.

**Limitations and future work.** Despite the promising results, DynaRend still relies on an external low-level motion planner to convert the predicted keyframe actions into executable motion sequences. An important direction for future work is to explore how to directly leverage the triplane representations for action sequence prediction, enabling more integrated and end-to-end robot control.

## Acknowledgments

This work was supported in part by the National Key Research and Development Project under Grant 2024YFB4708100, National Natural Science Foundation of China under Grants 62088102, U24A20325 and 12326608, and Key Research and Development Plan of Shaanxi Province under Grant 2024PT-ZCK-80.

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

# A  Implementation Details

**Architecture.** In this section, we detail the architecture of the reconstructive model, predictive model, and render head. Both the reconstructive and predictive models share the same Transformer architecture, which includes recent techniques such as SwiGLU [35], RoPE [37], and QK Norm [16]. To handle the 3D triplane representations, we utilize rotary position encoding inspired by M-RoPE [40]. The feature dimensions are split into three parts, corresponding to the three spatial dimensions of the triplane (*i.e.*, $x-y$, $x-z$, and $y-z$ planes). For each dimension, we apply RoPE encoding separately, which allows the model to effectively learn relative positional relationships across the three triplane planes.

The render head takes the triplane features, interpolated according to the sampled points, as input and outputs the corresponding density, RGB values, and semantic features. The render head is implemented as two MLP layers with residual connections. Each layer consists of a linear layer, followed by layer normalization and a ReLU activation followed by a linear layer. This architecture enables efficient feature processing and representation learning for both the reconstruction and prediction tasks, with the same render head shared across both tasks.

| Hyperparameter | Value |
| --- | --- |
| triplane resolution | $16\times16\times16$ |
| transformer depth | 8 |
| transformer width | 768 |
| attention heads | 12 |
| MLP ratio | 4.0 |
| render head width | 768 |
| render head layers | 2 |
| batch size | 256 |
| ray batch size | 32 |
| optimizer | AdamW |
| learning rate | 0.0001 |
| corase sampled points | 128 |
| fine depth sampled points | 64 |
| fine uniform sampled points | 64 |
| $\lambda_{pred}$ | 1.0 |
| $\lambda_{recon}$ | 1.0 |
| $\lambda_{rgb}$ | 1.0 |
| $\lambda_{sem}$ | 0.1 |
| $\lambda_{depth}$ | 0.01 |

Table 5: **Hyperparameters.**

Additionally, we follow GNFactor [46] by adopting a coarse-to-fine hierarchical structure for rendering. In the "fine" network, we apply depth-guided sampling to refine the predictions, improving the model's ability to reconstruct and predict scene features, especially at finer levels of detail.

**Target view augmentation.** In real-world setups, the limited and often fixed camera viewpoints pose a challenge for rendering-based pretraining. To address this limitation, we leverage a pretrained visual-conditioned multi-view diffusion model to generate novel target views as additional supervision. Specifically, we utilize See3D [27], a large-scale pretrained multi-view video diffusion model. Given input RGB-D images for a target frame, we first reconstruct the scene's

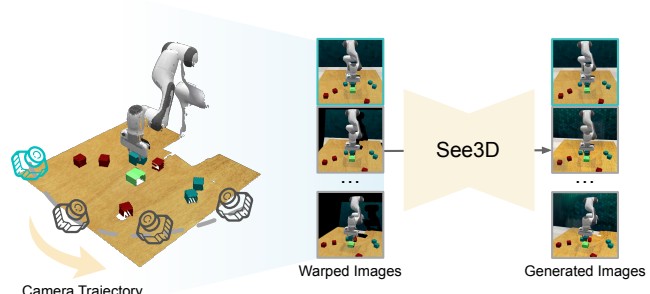

Figure 6: **Target view augmentation pipeline.**

point cloud. We then select an input viewpoint as the base view, from which we sample a new viewpoint with a random angular offset on a sphere surrounding the center of the scene. The angular offset is restricted within a maximum range of $\pm30$ degrees, as we observe that the quality of the synthesized novel view images decreases as the offset angle becomes larger. We generate a camera trajectory consisting of 25 frames, where each frame corresponds to a camera pose interpolated between the original and new viewpoints. We then use point cloud rendering to project the reconstructed point cloud onto each frame of the trajectory, producing warped images and corresponding masks. These warped images and masks, along with the original view, are fed into See3D, which generates high-fidelity novel-view images. Due to the time required for generating these trajectories, we augment only the keyframe viewpoints. For each keyframe, we randomly sample four distinct camera trajectories, generating a total of 100 novel view images per keyframe. In both simulation and real-world experiments, we maintain fixed camera setups and viewpoints, with the view augmentation process described above applied consistently.

**Hyperparameters.** We present the hyperparameters used in DynaRend as shown in Tab. 5.

# B Simulation Experiment Details

**RLBench-18.** For the 18-task setting, we follow PerAct [36] and select 18 RLBench tasks that involve at least two or more variations in object types, spatial arrangements, or instructions. This curated subset is designed to evaluate the multi-task generalization capabilities of different agents under diverse and realistic conditions. Detailed descriptions of each task can be found in Tab. 6.

| Task | Variation Type | # Variations | Avg. Keyframes | Language Template |
|------|---------------|-------------|----------------|-------------------|
| open drawer | placement | 3 | 3.0 | "open the __ drawer" |
| slide block | color | 4 | 4.7 | "slide the block to __ target" |
| sweep to dustpan | size | 2 | 4.6 | "sweep dirt to the __ dustpan" |
| meat off grill | category | 2 | 5.0 | "take the __ off the grill" |
| turn tap | placement | 2 | 2.0 | "turn __ tap" |
| put in drawer | placement | 3 | 12.0 | "put the item in the __ drawer" |
| close jar | color | 20 | 6.0 | "close the __ jar" |
| drag stick | color | 20 | 6.0 | "use the stick to drag the cube onto the __ target" |
| stack blocks | color,count | 60 | 14.6 | "stack __ __ blocks" |
| screw bulb | color | 20 | 7.0 | "screw in the __ light bulb" |
| put in safe | placement | 3 | 5.0 | "put the money away in the safe on the __ shelf" |
| place wine | placement | 3 | 5.0 | "stack the wine bottle to the __ of the rack" |
| put in cupboard | category | 9 | 5.0 | "put the __ in the cupboard" |
| sort shape | shape | 5 | 5.0 | "put the __ in the shape sorter" |
| push buttons | color | 50 | 3.8 | "push the __ button, [then the __ button]" |
| insert peg | color | 20 | 5.0 | "put the ring on the __ spoke" |
| stack cups | color | 20 | 10.0 | "stack the other cups on top of the __ cup" |
| place cups | count | 3 | 11.5 | "place __ cups on the cup holder" |

Table 6: **Details on 18 RLBench tasks.**

**RLBench-71.** For large-scale multi-task evaluation, we follow SPA [47] and divide all executable tasks in RLBench into two groups, consisting of 35 and 36 tasks respectively. For each group, we train a language-conditioned multi-task agent. All RVT-2 [12] baselines are implemented using the same two-stage architecture, differing only in the choice of pretrained vision encoder. All other components and training hyperparameters remain consistent with the original RVT-2 setup. Reported results are taken directly from SPA [47].

The 35 tasks in Group 1 include: `basketball in hoop, put rubbish in bin, meat off grill, meat on grill, slide block to target, reach and drag, take frame off hanger, water plants, hang frame on hanger, wipe desk, stack blocks, reach target, push button, lamp on, toilet seat down, close laptop lid, open box, open drawer, pick up cup, turn tap, take usb out of computer, play jenga, insert onto square peg, take umbrella out of umbrella stand, insert usb in computer, straighten rope, turn oven on, change clock, close microwave, close fridge, close grill, open grill, unplug charger, press switch, take money out safe`.

The 36 tasks in Group 2 include: The 36 tasks in Group 2 include: `change channel, tv on, push buttons, stack wine, scoop with spatula, place hanger on rack, move hanger, sweep to dustpan, take plate off colored dish rack, screw nail, take shoes out of box, slide cabinet open and place cups, lamp off, pick and lift, take lid off saucepan, close drawer, close box, phone on base, toilet seat up, put books on bookshelf, beat the buzz, stack cups, put knife on chopping board, place shape in shape sorter, take toilet roll off stand, put umbrella in umbrella stand, setup checkers, open window, open wine bottle, open microwave, put money in safe, open door, close door, open fridge, open oven, plug charger in power supply`.

**Colosseum.** Colosseum [32] is a recently proposed benchmark designed to evaluate the generalization capabilities of robotic manipulation policies under diverse domain shifts. It consists of 20 manipulation tasks drawn from the RLBench suite, including: `basketball in hoop, close box, close laptop lid, empty dishwasher, get ice from fridge, hockey, meat on grill, move hanger, wipe desk, open drawer, slide block to target, reach and drag, put money in safe, place wine at rack location, insert onto square peg, stack cups, turn oven on, straighten rope, setup chess, scoop with spatula`.

Colosseum defines 14 types of perturbation factors to simulate real-world domain shifts. In our evaluation, we follow the standard setting and apply 12 of these perturbation factors to all 20 tasks

to systematically assess the robustness and generalization ability of learned policies under unseen conditions. The 12 perturbations can be categorized into three groups:

- **Manipulation Object (MO) Perturbation:** The manipulation object is the item directly manipulated by the robot, such as the `wine bottle` in `place wine at rack location`. Manipulation object perturbations include variations in color, texture, and size.
- **Receiver Object (RO) Perturbation:** The receiver object is task-relevant but not directly manipulated, such as the `rack` in the same task. Receiver object perturbations also include changes in color, texture, and size.
- **Background Perturbation:** These factors affect the overall scene without modifying task-relevant objects. They include changes to `light color`, `table color`, `table texture`, `background texture`, `camera pose`, and the addition of `distractor objects`.

Tab. 7 summarizes the applied perturbations for each task. For more implementation details and perturbation configurations, please refer to Colosseum [32].

| Variation | MO | RO | MO Color | MO Size | MO Texture | RO Color | RO Size | RO Texture | Object Mass |
|---|---|---|---|---|---|---|---|---|---|
| | - | - | discrete | continuous | discrete | discrete | continuous | discrete | continuous |
| basketball in hoop | ball | hoop | 20 | [0.75,1.25] | 213 | 20 | [0.75,1.15] | - | - |
| close box | box | - | 20 | [0.75,1.15] | - | - | - | - | - |
| close laptop lid | laptop | - | 20 | [0.75,1.00] | - | - | - | - | - |
| empty dishwasher | dishwasher | plate | 20 | [0.80,1.00] | - | 20 | [0.80,1.00] | 213 | - |
| get ice from fridge | cup | fridge | 20 | [0.75,1.25] | 213 | 20 | [0.75,1.00] | - | - |
| hockey | stick | ball | 20 | [0.95,1.05] | - | 20 | [0.75,1.25] | 213 | [0.1,0.5] |
| meat on grill | meat | grill | 20 | [0.65,1.15] | - | 20 | - | - | - |
| move hanger | hanger | pole | 20 | - | - | 20 | - | - | - |
| wipe desk | sponge | beans | 20 | [0.75,1.25] | 213 | 20 | - | - | [1.0,5.0] |
| open drawer | drawer | - | 20 | [0.75,1.00] | - | - | - | - | - |
| slide block to target | block | - | 20 | - | 213 | - | - | - | [1.0,15.0] |
| reach and drag | stick | block | 20 | [0.80,1.10] | 213 | 20 | [0.50,1.00] | 213 | [0.5,2.5] |
| put money in safe | money | safe | 20 | [0.50,1.00] | 213 | 20 | - | 213 | - |
| place wine at rack location | bottle | shelve | 20 | [0.85,1.15] | - | 20 | [0.85,1.15] | 213 | - |
| insert onto square peg | peg | spokes | 20 | [1.00,1.50] | - | 20 | [0.85,1.15] | 213 | - |
| stack cups | cups | - | 20 | [0.75,1.25] | 213 | - | - | - | - |
| turn oven on | knobs | - | 20 | [0.50,1.50] | - | - | - | - | - |
| straighten rope | rope | - | 20 | - | 213 | - | - | - | - |
| setup chess | chess pieces | board | 20 | [0.75,1.25] | 213 | 20 | - | - | - |
| scoop with spatula | spatula | block | 20 | [0.75,1.25] | 213 | 20 | [0.75,1.50] | 213 | [1.0,5.0] |

Table 7: **Summary of tasks and their perturbation factors.** For more details, check Colosseum [32].

## C Real-world Experiment Details

**Hardware Setup.** For the hardware setup, we use a Franka Research 3 robot arm equipped with a parallel gripper. Visual input is provided by two Orbbec Femto Bolt RGB-D cameras, positioned on either side of the robot arm. The cameras are calibrated using the `kalibr` package to determine the extrinsics between them, while `easy handeye` package is used to calibrate the extrinsics between the camera and the robot base-frame. The cameras provide RGB-D images at a resolution of 1280x720 at 30Hz. To prepare the images for model input, we resize the shortest edge to 256 and crop them to a resolution of 256x256.

| Task | Variation Type | # Variations | Avg. Keyframes | Language Template |
|---|---|---|---|---|
| stack cups | color | 5 | 11.5 | "stack __ cups on __ cup" |
| stack blocks | color | 5 | 12.1 | "stack __ blocks on __ block" |
| sort shape | placement | 1 | 5.4 | "put yellow circle in shape sorter" |
| close pot | placement | 1 | 5.7 | "put lid on pot" |
| put item in drawer | category | 4 | 8.3 | "put __ in drawer" |

Table 8: **Details on 5 real-world tasks.**

**Data Collection.** The real-world dataset is collected through human demonstrations. Specifically, for each task and scenario, a human demonstrator kinesthetically moves the robot arm to specify a series of end-effector poses. Afterward, the robot arm is reset to its initial position, and the demonstrator sequentially moves it to the specified poses. During this process, the camera streams and robot arm states are recorded, including end-effector position, joint positions, and joint velocities. For each task, we collect 30 human demonstrations to train the model.

| Task Name | No Variations | MO Color | RO Color | MO Texture | RO Texture | MO Size | RO Size | Light Color | Table Color | Table Texture | Distractor | Background Texture | Camera Pose |
|---|---|---|---|---|---|---|---|---|---|---|---|---|---|
| basketball in hoop | 100 | 96 | 100 | 96 | - | 100 | 100 | 100 | 100 | 100 | 40 | 100 | 100 |
| close box | 96 | 84 | - | - | - | 92 | - | 92 | 88 | 88 | 88 | 96 | 88 |
| close laptop lid | 100 | 100 | - | - | - | 0 | - | 100 | 100 | 96 | 68 | 100 | 96 |
| empty dishwasher | 0 | 0 | 0 | - | 0 | 0 | 0 | 0 | 0 | 0 | 0 | 0 | 0 |
| get ice from fridge | 92 | 88 | 96 | 88 | - | 80 | 72 | 96 | 100 | 96 | 64 | 92 | 100 |
| hockey | 32 | 36 | 36 | - | 20 | 44 | 32 | 28 | 36 | 20 | 28 | 28 | 32 |
| meat on grill | 96 | 100 | 100 | - | - | 100 | - | 100 | 100 | 100 | 96 | 96 | 100 |
| move hanger | 0 | 0 | 0 | - | - | - | - | 0 | 0 | 0 | 8 | 0 | 0 |
| wipe desk | 0 | 0 | - | 0 | - | 0 | - | 0 | 0 | 0 | 0 | 0 | 0 |
| open drawer | 84 | 68 | - | - | - | 68 | - | 68 | 72 | 60 | 64 | 68 | 68 |
| slide block to target | 100 | 100 | - | 100 | - | - | - | 100 | 100 | 100 | 100 | 100 | 100 |
| reach and drag | 100 | 96 | 96 | 100 | 100 | 100 | 52 | 100 | 96 | 96 | 60 | 100 | 96 |
| put money in safe | 56 | 80 | 4 | 60 | 64 | 68 | - | 64 | 60 | 68 | 60 | 76 | 68 |
| place wine at rack location | 88 | 80 | 92 | - | 80 | 52 | 84 | 80 | 84 | 88 | 96 | 88 | 88 |
| insert onto square peg | 8 | 4 | 12 | - | 28 | 16 | 12 | 16 | 0 | 16 | 0 | 8 | 8 |
| stack cups | 72 | 68 | - | 52 | - | 36 | - | 48 | 52 | 60 | 36 | 72 | 56 |
| turn oven on | 76 | 100 | - | - | - | 56 | - | 92 | 92 | 96 | 100 | 100 | 88 |
| straighten rope | 80 | 64 | - | 84 | - | - | - | 64 | 72 | 68 | 24 | 60 | 80 |
| setup chess | 20 | 0 | 4 | 16 | - | 16 | - | 16 | 24 | 24 | 12 | 8 | 12 |
| scoop with spatula | 84 | 56 | 96 | 80 | 88 | 96 | 60 | 16 | 88 | 96 | 56 | 88 | 88 |
| **average** | 64.2 | 61.0 | 53.0 | 67.6 | 54.2 | 54.3 | 51.5 | 59.0 | 63.2 | 63.6 | 50.0 | 64.0 | 63.4 |

Table 10: **All results on Colosseum benchmark.**

**Keyframe Discovery.** Following the approach of PerAct [36], we use heuristic rules to identify keyframes from the collected demonstrations. A frame is considered a keyframe if: 1) The joint velocities are near zero, and 2) The gripper open state has not changed.

**Execution.** For robot control and motion planning, we use the Franka ROS and MoveIt with RRT-Connect as default planner.

# D  Additional Results

**Additional baselines.** We include additional baselines in Tab. 9 to further compare the performance of DynaRend with existing pretraining methods on the RLBench-18 tasks under the *in-domain* setting, where models are pretrained directly on RLBench-18 data.

| Method | Pretrain Data | Avg. S.R. |
|---|---|---|
| RVT [13] | None | 62.9 |
| MVP [43] | RLBench | 64.9 |
| 3D-MVP [33] | RLBench | 67.5 |
| DynaRend | RLBench | **83.2** |

Table 9: **Results on 18 RLBench tasks.**

**Detailed results.** We present the detailed results of all tasks in Tab. 10 and Tab. 11.

# E  Visualizations

In Fig. 8 and Fig. 10, we present the results of target view augmentation in both the simulator and real-world environments. Fig. 7 and Fig. 9 showcase the rendering results in the simulator and real-world settings, respectively.

# F  Additional Qualitative Analysis

In the supplementary materials, we provide episode rollouts of several representative tasks in the simulator and real-world demos in the attached videos.

| Method | RVT | MoCov3 | MAE | DINOv2 | CLIP | EVA | InternViT | MVP | VC-1 | SPA | **DynaRend** |
|---|---|---|---|---|---|---|---|---|---|---|---|
| *Group 1* | | | | | | | | | | | |
| basket ball in hoop | 100 | 100 | 100 | 100 | 100 | 100 | 100 | 100 | 100 | 100 | 100 |
| put rubbish in bin | 96 | 100 | 100 | 96 | 96 | 96 | 100 | 96 | 100 | 100 | 100 |
| meat off grill | 100 | 100 | 100 | 100 | 100 | 100 | 100 | 100 | 100 | 100 | 100 |
| meat on grill | 88 | 80 | 76 | 76 | 68 | 80 | 72 | 68 | 76 | 80 | 92 |
| slide block to target | 8 | 0 | 84 | 96 | 24 | 4 | 0 | 100 | 100 | 4 | 100 |
| reach and drag | 100 | 100 | 96 | 88 | 100 | 96 | 100 | 96 | 100 | 100 | 100 |
| take frame off hanger | 96 | 88 | 88 | 92 | 88 | 84 | 84 | 88 | 88 | 96 | 96 |
| water plants | 48 | 64 | 60 | 28 | 64 | 60 | 44 | 52 | 60 | 68 | 56 |
| hang frame on hanger | 0 | 8 | 4 | 0 | 4 | 8 | 8 | 12 | 4 | 4 | 24 |
| wipe desk | 0 | 0 | 0 | 0 | 0 | 0 | 0 | 0 | 0 | 0 | 0 |
| stack blocks | 56 | 60 | 72 | 72 | 68 | 56 | 60 | 84 | 68 | 68 | 80 |
| reach target | 96 | 60 | 96 | 88 | 100 | 96 | 80 | 92 | 96 | 92 | 100 |
| push button | 92 | 100 | 100 | 100 | 100 | 100 | 100 | 100 | 100 | 100 | 100 |
| lamp on | 60 | 88 | 68 | 84 | 88 | 52 | 80 | 28 | 88 | 64 | 100 |
| toilet seat down | 100 | 100 | 100 | 100 | 100 | 100 | 100 | 96 | 96 | 100 | 96 |
| close laptop lid | 84 | 96 | 96 | 96 | 96 | 84 | 80 | 80 | 96 | 100 | 96 |
| open drwaer | 92 | 88 | 96 | 92 | 100 | 88 | 88 | 92 | 96 | 96 | 96 |
| pick up cup | 88 | 92 | 92 | 88 | 96 | 96 | 88 | 96 | 96 | 96 | 96 |
| turn tap | 88 | 88 | 84 | 84 | 96 | 88 | 92 | 96 | 100 | 100 | 96 |
| take usb out of computer | 100 | 100 | 100 | 100 | 100 | 100 | 100 | 100 | 88 | 100 | 92 |
| play jenga | 100 | 96 | 96 | 96 | 100 | 96 | 100 | 96 | 96 | 96 | 100 |
| insert onto square peg | 8 | 28 | 84 | 80 | 44 | 88 | 40 | 64 | 92 | 84 | 24 |
| take umbrella out of umbrella stand | 92 | 92 | 100 | 100 | 92 | 100 | 96 | 100 | 100 | 100 | 100 |
| insert usb in computer | 8 | 12 | 20 | 20 | 24 | 24 | 20 | 16 | 8 | 64 | 12 |
| straighten rope | 56 | 56 | 44 | 72 | 80 | 48 | 72 | 52 | 60 | 84 | 56 |
| turn oven on | 92 | 96 | 96 | 96 | 96 | 96 | 96 | 100 | 100 | 100 | 96 |
| change clock | 64 | 64 | 68 | 48 | 68 | 64 | 72 | 64 | 60 | 68 | 68 |
| close microwave | 100 | 100 | 100 | 100 | 100 | 100 | 100 | 100 | 100 | 100 | 100 |
| close fridge | 92 | 80 | 92 | 92 | 88 | 92 | 96 | 88 | 92 | 100 | 96 |
| close grill | 92 | 96 | 96 | 96 | 96 | 96 | 96 | 100 | 100 | 96 | 96 |
| open grill | 76 | 100 | 100 | 100 | 100 | 100 | 100 | 96 | 100 | 100 | 96 |
| unplug charger | 48 | 44 | 32 | 48 | 36 | 48 | 40 | 40 | 44 | 44 | 96 |
| press switch | 84 | 92 | 92 | 88 | 72 | 76 | 84 | 76 | 88 | 92 | 76 |
| take money out safe | 80 | 100 | 96 | 100 | 100 | 100 | 100 | 100 | 100 | 100 | 100 |
| *Group 2* | | | | | | | | | | | |
| change channel | 0 | 0 | 8 | 4 | 0 | 0 | 4 | 0 | 0 | 4 | 100 |
| tv on | 0 | 4 | 8 | 0 | 4 | 4 | 8 | 4 | 4 | 8 | 100 |
| push buttons | 0 | 12 | 4 | 4 | 0 | 0 | 0 | 0 | 12 | 4 | 96 |
| stack wine | 12 | 12 | 16 | 40 | 4 | 12 | 0 | 28 | 8 | 28 | 20 |
| scoop with spatula | 0 | 0 | 0 | 0 | 0 | 0 | 0 | 0 | 0 | 0 | 92 |
| place hanger on rack | 0 | 0 | 0 | 0 | 0 | 0 | 0 | 0 | 0 | 0 | 100 |
| move hanger | 72 | 0 | 0 | 0 | 0 | 0 | 0 | 0 | 0 | 0 | 88 |
| sweep to dustpan | 48 | 92 | 96 | 96 | 96 | 92 | 100 | 100 | 88 | 96 | 100 |
| take plate off colored dish rack | 92 | 96 | 100 | 96 | 92 | 84 | 96 | 88 | 92 | 96 | 84 |
| screw nail | 32 | 52 | 36 | 36 | 36 | 36 | 52 | 32 | 32 | 48 | 56 |
| take shoes out of box | 4 | 20 | 28 | 24 | 36 | 40 | 12 | 32 | 36 | 36 | 8 |
| slide cabinet open and place cups | 0 | 0 | 0 | 0 | 0 | 0 | 4 | 0 | 0 | 4 | 0 |
| lamp off | 96 | 100 | 96 | 96 | 100 | 96 | 96 | 100 | 100 | 100 | 100 |
| pick and lift | 72 | 88 | 96 | 92 | 96 | 92 | 80 | 96 | 96 | 96 | 88 |
| take lid off saucepan | 100 | 100 | 100 | 100 | 100 | 100 | 100 | 100 | 100 | 100 | 100 |
| close drawer | 100 | 100 | 100 | 100 | 100 | 96 | 100 | 100 | 100 | 100 | 100 |
| close box | 96 | 92 | 92 | 96 | 96 | 100 | 96 | 100 | 96 | 100 | 88 |
| phone on base | 100 | 100 | 100 | 100 | 100 | 100 | 96 | 100 | 100 | 100 | 100 |
| toilet seat up | 72 | 80 | 88 | 100 | 88 | 88 | 80 | 88 | 92 | 96 | 24 |
| put books on bookshelf | 28 | 12 | 24 | 24 | 28 | 28 | 20 | 20 | 28 | 16 | 28 |
| beat the buzz | 88 | 84 | 92 | 96 | 88 | 84 | 88 | 88 | 88 | 100 | 100 |
| stack cups | 24 | 40 | 56 | 52 | 52 | 48 | 56 | 64 | 68 | 64 | 68 |
| put knife on chopping board | 80 | 72 | 76 | 68 | 72 | 80 | 88 | 80 | 76 | 80 | 72 |
| place shape in shape sorter | 12 | 20 | 36 | 32 | 28 | 36 | 20 | 44 | 36 | 56 | 48 |
| take toilet roll off stand | 100 | 100 | 92 | 76 | 96 | 92 | 88 | 84 | 92 | 96 | 84 |
| put umbrella in umbrella stand | 20 | 8 | 0 | 12 | 12 | 0 | 4 | 12 | 8 | 12 | 12 |
| setup checkers | 44 | 76 | 80 | 68 | 68 | 88 | 92 | 92 | 80 | 80 | 92 |
| open window | 92 | 96 | 96 | 100 | 100 | 96 | 100 | 96 | 100 | 100 | 92 |
| open wine bottle | 96 | 80 | 100 | 88 | 92 | 92 | 88 | 96 | 88 | 88 | 100 |
| open microwave | 72 | 100 | 100 | 88 | 96 | 100 | 80 | 96 | 100 | 100 | 84 |
| put money in safe | 100 | 96 | 100 | 88 | 92 | 100 | 96 | 100 | 100 | 100 | 96 |
| open door | 88 | 100 | 96 | 96 | 96 | 96 | 96 | 84 | 96 | 96 | 92 |
| close door | 12 | 32 | 68 | 56 | 60 | 80 | 20 | 24 | 20 | 60 | 0 |
| open fridge | 24 | 44 | 52 | 48 | 44 | 36 | 64 | 52 | 32 | 64 | 68 |
| open oven | 0 | 8 | 4 | 12 | 8 | 4 | 20 | 4 | 4 | 16 | 80 |
| plug charger in power supply | 40 | 32 | 36 | 32 | 24 | 44 | 36 | 24 | 32 | 60 | 28 |

Table 11: **All results on 71 RLBench tasks.**

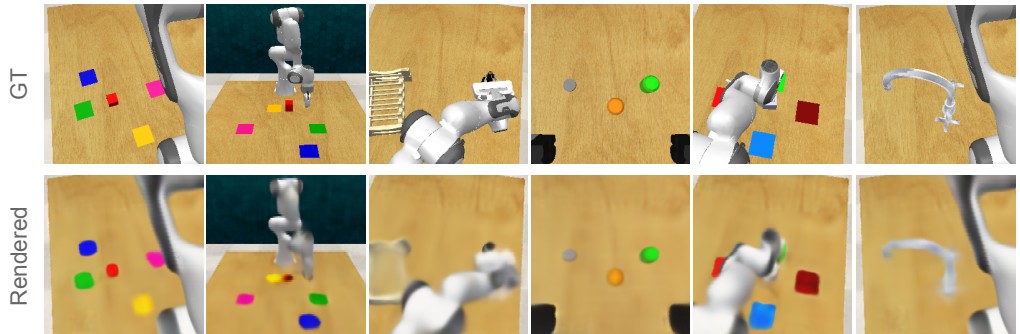

Figure 7: **Visualization of rendered results in simulation.**

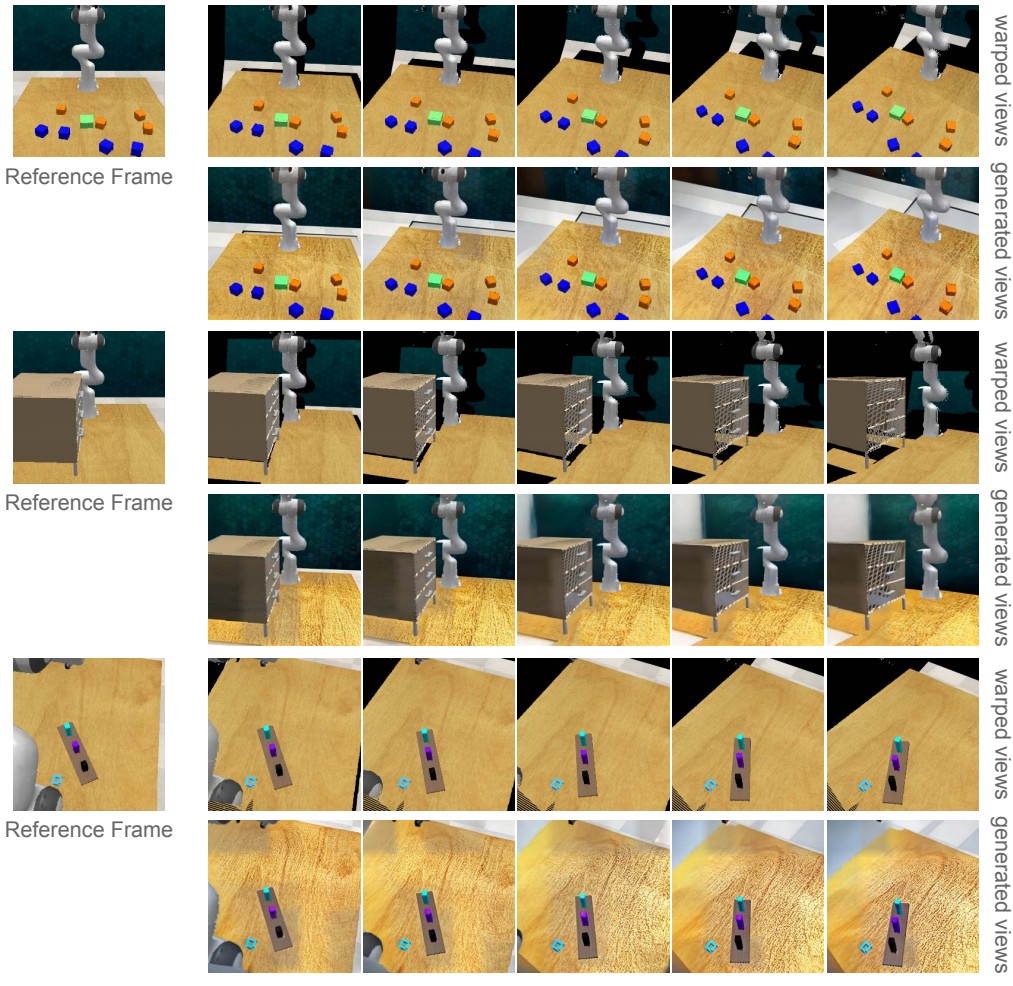

Figure 8: **Visualization of target view synthesis in simulation.**

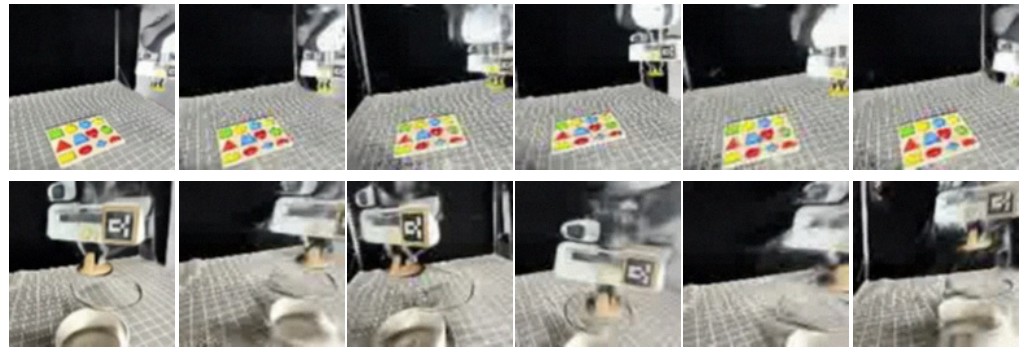

Figure 9: **Visualization of rendered results in the real world.**

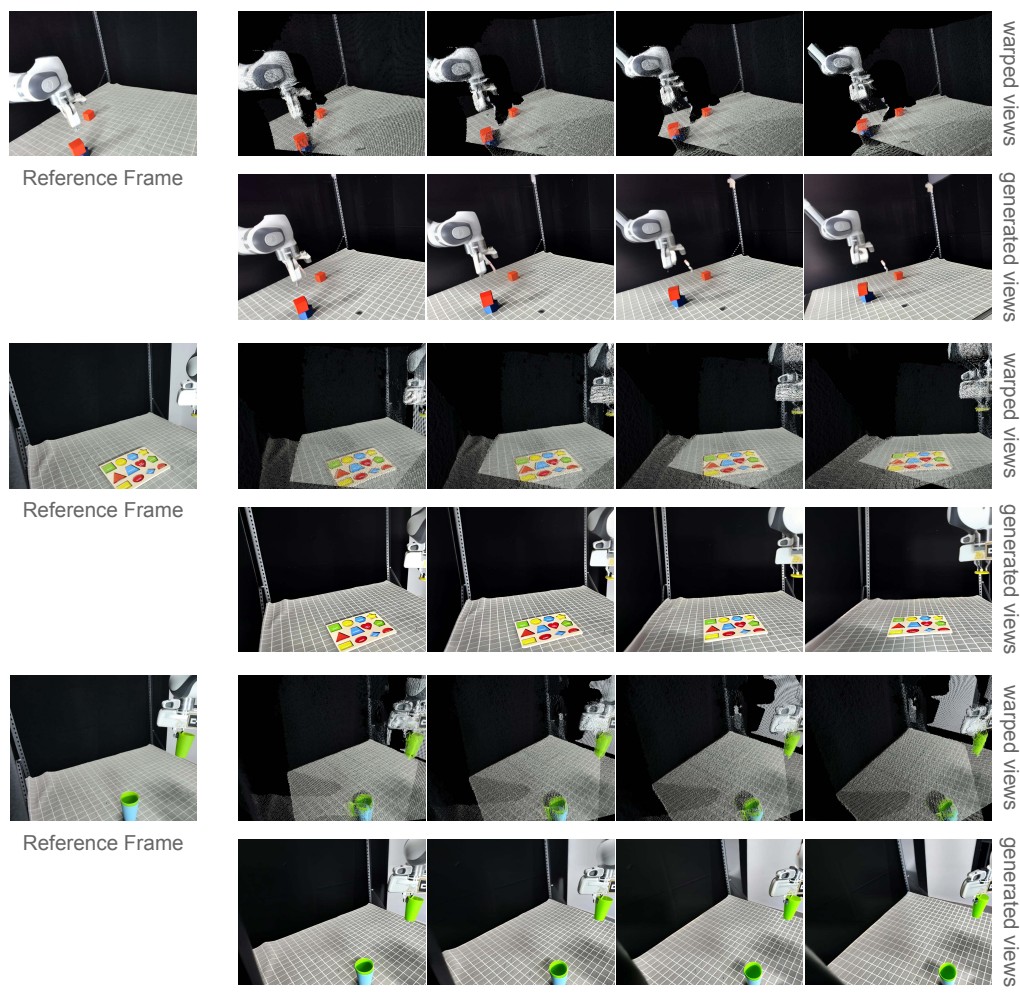

Figure 10: **Visualization of target view synthesis in the real world.**

