# OpenReview forum: "DynaRend: Learning 3D Dynamics via Masked Future Rendering for Robotic Manipulation"
_NeurIPS.cc/2025/Conference — NeurIPS 2025 poster_

### Official Review · Reviewer_ovuc · 2025-06-29

**Clarity:** 3
**Significance:** 2
**Originality:** 2
**Rating:** 2
**Confidence:** 3

**Summary:**

DynaRend is a self‑supervised framework that masks and predicts future triplane volumes inside a differentiable renderer, yielding 3D‑ and dynamics‑aware features. When reused as a frozen/finetuned encoder, these features deliver large gains on RLBench and several real‑world manipulation setups.

**Questions:**

The questions are the same as the weakness outlined above.

**Ethical Concerns:**

["NO or VERY MINOR ethics concerns only"]

**Limitations:**

yes

**Quality:**

2

**Strengths And Weaknesses:**

Strengths:

1. Comprehensive evaluation across simulation, robustness benchmarks, and five real‑robot tasks with strong absolute gains.
2. Clear exposition with a step‑by‑step pipeline diagram, well‑structured text, and thorough ablations.

Weakness:

1. Performance doesn’t seem that strong. The gap between the proposed method and other baselines is slight.
2. I think the pretrained encoder will overfit to the given dataset distribution. I doubt its generalizability in terms of object types or scenes. In robotics, there are many cases where the robot has to address out-of-distribution cases, but this method couldn’t address this case.
3. This method requires a calibrated multi-view dataset, which is infeasible in most real-world cases. The author said this makes it practical for scalable deployment in real-world setups, but I think the opposite. The usual/practical robot applications come with 1-2 cameras, which is much smaller than the required camera viewpoints in this work. Furthermore, it hasn’t even enabled the robot to work with a single view during downstream deployment.
4. No action is engaged in the predictive network, which fails to capture a dynamics-aware feature. Even though the current triplane feature is the same, the next keyframe’s triplane feature would be different according to what action the robot takes. But the current objective in the prediction phase does not consider this, which is contradictory to the author’s arguments.
5. Techniques are incrementally built upon the existing method, such as SNeRL, RVT, etc. I think this method lacks technical novelty.

---

> ### Author Rebuttal · Authors · 2025-07-31
>
> Dear Reviewer ovuc,
>
> Many thanks for your valuable comments and questions. We hereby address your concerns as follows.
>
> ---
>
> > ### \[W1\]: Limited performance gain.
>
> - RVT-2 attains high performance partly due to its **two-stage prediction strategy**, while our current framework adopts a **simpler single-stage design**. For a fair comparison, our results should be viewed against other single-stage baselines. Even so, our method surpasses RVT-2 under comparable conditions, indicating strong effectiveness. Importantly, our architecture is fully **compatible with a two-stage extension** like RVT-2, which could further increase performance if desired. To verify this, we additionally implemented a two-stage variant of our framework and evaluated it on 18 RLBench tasks, where it achieved clear improvements over the previous SOTA.
>
>     |      | Avg. S.R. $\uparrow$ |
>     | :----      | :-------: |
>     | RVT        |     62.9      |
>     | RVT-2      |     81.4      |
>     | DynaRend   |     83.2      |
>     | DynaRend-2 |     **87.8**      |
>
> - The RLBench 18-task suite contains relatively simple tasks, where many methods already approach saturation, leaving limited headroom for large absolute gains. Following prior work, we train on the PerAct-generated dataset (100 demos/task), which inevitably omits some task variations and constrains achievable performance. To better measure generalization, we evaluate on **all executable RLBench tasks(71 tasks)** and compare against RVT-2 variants with different pretraining strategies. In this broader setting, our method improves success rate by 8.2%, underscoring its robustness beyond the standard benchmark.
>
> > ### \[W2\]: Generalizability in out-of-distribution cases.
>
> Overfitting in imitation learning often arises when policies memorize scene layouts or latch onto spurious correlations, especially given noisy action labels. This results in poor transfer to novel conditions. Our approach addresses this by using self-supervised pretraining via 3D future prediction, enabling the model to capture how the environment evolves and to develop dynamics-aware features that support inverse dynamics reasoning. These features mitigate reliance on brittle visual-to-action mappings.
>
> To evaluate generalization, we adopt the Colosseum[1] benchmark, which is explicitly designed to evaluate robot policies under 12 distinct types of  out-of-distribution environment perturbations, such as object size, unseen distractors, lighting shifts, novel textures and so on. Each perturbation type targets a different failure mode in manipulation policies, making the benchmark a comprehensive proxy for real-world distribution shifts. Across all perturbation types, our method achieves an average success rate improvement of 18.2% over the previous SOTA, 3D-MVP (in Fig. 4).
>
> Additionally, To further validate robustness in real-world deployment, we conduct experiments under a challenging setting where novel, unseen distractor objects are introduced into the workspace during policy execution. Under this setting, the success rate of RVT-2 drops sharply by 57%, indicating a strong reliance on memorized scene layouts or specific visual cues. In contrast, our policy—initialized with 3D-aware masked future prediction pretraining—shows only a 21% drop in success rate.
>
> These results confirm stronger robustness to both synthetic and real-world distribution shifts. In future work, we plan to pretrain on large-scale multi-view datasets (e.g., ScanNet[2], DROID[3]) to further expand coverage of real-world variability.
>
> > ### \[W3\]: Reliance on extensive camera viewpoints.
>
> We appreciate the reviewer for the comment but respectfully note that our method does not require many cameras for deployment. In real-world experiments, we use only **two fixed cameras** and outperform RVT-2 under the same configuration. Four cameras are used only in simulation for fair comparison with prior work. Our method also remains effective with single-view setups. As shown below, performance drops only slightly when using a single camera.
>
> | | \# Cameras | Put Item in Drawer | Stack Blocks | Sort Shape | Close Pot | Stack Cups | Avg.S.R. $\uparrow$ |
> | :- | :-: | :-: | :-: | :-: | :-: | :-: | :-: |
> | RVT-2 | 1 | 50 | 60 | 10 | 60 | 5 | 35 |
> | RVT-2 | 2 | 45 | 60 | 10 | 60 | 15 | 37 |
> | DynaRend | 1 | 65 | 55 | 25| 80 | 45 | 54 |
> | DynaRend | 2 | 65 | 60 | 35 | 85 | 40 | **57** |
>
> Table: Real-world performance.
>
>
> | | \# Cameras | Put Item in Drawer | Stack Blocks | Sort Shape | Close Pot | Stack Cups | Avg.S.R. $\uparrow$ |
> | :- | :-: | :-: | :-: | :-: | :-: | :-: | :-: |
> | RVT-2 | 1 | 15 | 5 | 10 | 40 | 5 | 15 |
> | RVT-2 | 2 | 15 | 10 | 10 | 45 | 0 | 16 |
> | DynaRend | 1 | 45 | 50 | 25 | 65 | 25 | 42 |
> | DynaRend | 2 | 55 | 55 | 25 | 65 | 25 | **45** |
>
> Table: Real-world performance with additional distractors.
>
>
> Crucially, our claim of practicality refers specifically to prior rendering-based pretraining methods (e.g., GNFactor), which rely on rendering large amounts of novel views in simulation—a process that is infeasible in the real world. Instead, we leverage a visual-conditioned diffusion model to generate novel views from limited input views, which substantially reduces the need for dense multi-view data and makes rendering-based pretraining applicable in real-world settings.
>
> > ### \[W4\]: Action engagement in predictive network.
>
> While we do not feed explicit actions into the predictive network, it is designed to learn dynamics-aware features directly from visual evolution. Introducing action inputs during pretraining risks **information leakage**, potentially causing the model to **overfit to specific action-conditioned cues** rather than learning transferable spatial-temporal structure.
>
> Robotic actions are inherently multi-modal; different actions can yield diverse future states. To control this uncertainty, we predict semantically **meaningful keyframes** (following PerAct) rather than fixed-interval future frames. These keyframes correspond to **stable intermediate states** (e.g., pre-grasp and post-grasp), enabling the network to focus on informative state transitions that align with decision points, rather than noisy intermediate frames.
>
> > ### \[W5\]: Novelty of DynaRend.
>
> 1. DynaRend introduces a new paradigm for rendering-based representation learning in robotic manipulation by **jointly modeling spatial geometry and future dynamics** within an explicit 3D triplane representation. Unlike prior approaches (e.g., SNeRL, RVT) that focus solely on static scene representation, DynaRend **couples masked reconstruction with future prediction through differentiable volumetric rendering**. This 3D-aware future-prediction objective is directly aligned with the demands of manipulation tasks, where reasoning about how the scene will change after an action is as critical as perceiving its current structure.
>
> 2. DynaRend integrates visual-conditioned diffusion-based view synthesis (See3D) into the rendering-based pretraining pipeline. This design enables high-quality novel-view generation from **a minimal number of real-world cameras**, eliminating the dependence on dense multi-view rigs or simulator-rendered images that limit prior methods’ applicability outside simulation. By reducing the hardware and data requirements, DynaRend **makes 3D rendering-based pretraining practically deployable** in real-world robotic systems.
>
> 3. Extensive evaluations on RLBench, the Colosseum benchmark (12 perturbation types), and six diverse real-world manipulation tasks consistently show that our approach yields **higher success rates and substantially improved robustness** to appearance, geometry, and viewpoint shifts compared to state-of-the-art baselines. Notably, on Colosseum, DynaRend improves over the previous best (3D-MVP) by 18.2%, and in real-world unseen distractor tests, our policy’s performance drops by only 21%, compared to a 57% drop for RVT-2. These results confirm that the combination of **3D-aware masked future prediction** and **diffusion-based view augmentation** provides a meaningful and transferable advance in representation learning for robot manipulation.
>
> ---
>
> Thank you again for your constructive and detailed feedback. We hope our responses have addressed your concerns, and we will incorporate the above clarifications and improvements in the final version. Please let us know if you have further questions — we will be actively available throughout the rebuttal period and would greatly appreciate your reconsideration of our work.
>
> > references:\
> [1] THE COLOSSEUM: A Benchmark for Evaluating Generalization for Robotic Manipulation, 2024.\
> [2] DROID: A Large-Scale In-the-Wild Robot Manipulation Dataset, 2024.\
> [3] Video Prediction Policy: A Generalist Robot Policy with Predictive Visual Representations, 2025.

---

> > ### Comment · Reviewer_ovuc · 2025-08-05
> >
> > Thank you to the authors for their clarification. However, I still have some remaining questions.
> >
> > 1. I remain unconvinced that adding a 3D future-prediction objective alone necessarily yields better transfer to truly novel objects or distracting clutter. Could you quantify how different the evaluation objects (and distractors) are from the training set in terms of shape, texture, and required grasp strategies? For instance, within the same category, an object that demands a significantly different grasp pose may still break the policy. If performance degrades under such intra-class variations, the claimed generalizability is limited. Moreover, because your pipeline relies on an extensive 3D pre-training setup, would each new task require a similarly heavy data-collection and rendering stage? Please clarify the practical burden this imposes for real-world deployment scenarios.
> >
> > 2. My earlier question about generalization also encompassed viewpoint changes. Figure 4 provides only a brief glimpse, and there is no dedicated experiment demonstrating true view-invariance. Could you present results where camera poses differ substantially from those seen during training (e.g., elevated angles, lateral shifts), or provide ablation studies isolating viewpoint robustness? (If you had already done some related experiments. I don't mean you should do new experiments in the rebuttal period.) Without such evidence, it is difficult to conclude that the learned representation is genuinely view-invariant rather than memorizing specific camera configurations.

---

> > > ### Author Response · Authors · 2025-08-05
> > >
> > > Dear Reviewer ovuc,
> > >
> > > 1. We appreciate the reviewer’s follow-up questions and the opportunity to clarify the scope of our generalization evaluation and the practical requirements of our approach.
> > >
> > >    - **Scope of Generalization.** In robotic manipulation, generalization spans multiple aspects, including **visual/physical generalization** (robustness to changes in object appearance, scene layout, lighting, and distractors) and **behavior generalization** (transferring to entirely different object categories or tasks that require qualitatively new manipulation skills). Our evaluation focuses on visual/physical generalization, i.e., whether a policy remains robust under variations in object pose, background, color, texture, size, and the presence of distractors. Generalization to completely novel objects requiring substantially different grasp strategies falls into the domain of behavior generalization, which is beyond the scope of standard behavior cloning. The central aim of our work is to develop a unified multi-task robotic policy learning framework that, via behavior cloning, acquires manipulation skills robust to visual/physical perturbations.
> > >
> > >    - **Role of Future Prediction.** Incorporating a 3D future-prediction objective encourages the model to focus on the relationship between actions and visual observations, implicitly learning inverse dynamics. This leads to representations that are more robust to visual perturbations and clutter, as supported by our results in both Colosseum and real-world evaluations.
> > >
> > >    - **Practical Deployment Burden.** In behavior cloning, new tasks inherently require collecting demonstrations, which is a common assumption in prior works. Our pretraining stage uses the same demonstration dataset as policy training but without action labels, combined with our view augmentation strategy. As a result, no additional data collection or extra camera viewpoints are required for pretraining. This means that our approach introduces no extra data collection overhead beyond standard BC pipelines, and does not impose a heavier deployment burden.
> > >
> > >    - **Quantifying Object and Distractor Variation.** Following the Colosseum benchmark protocol, the variations include:
> > >
> > >         1\) Spatial variation: random placement within the workspace bounds.
> > >
> > >         2\) Texture variation: 213 randomized textures.
> > >
> > >         3\) Color variation: 20 randomized colors.
> > >
> > >         4\) Size variation: randomized scaling within predefined ranges.
> > >
> > >         5\) Distractors: 78 objects from the YCB Object Dataset, placed in the scene.
> > >
> > >         Further details can be found in the Colosseum benchmark paper.
> > >
> > > 2. Regarding viewpoint generalization, we follow the protocol defined in the Colosseum benchmark. Specifically, during evaluation we introduce random perturbations to the front, left, and right cameras (excluding the wrist camera) by altering both their positions and orientations. For position changes, the offsets are uniformly sampled from −0.1 to 0.1 along each axis. For orientation changes, the perturbations are uniformly sampled Euler angles from −0.05 to 0.05. Further implementation details can be found in the Colosseum benchmark paper.
> > >
> > >     While Figure 4 already reports the impact of different Colosseum perturbations on policy performance including the camera pose variation setting. For clarity, we provide below the exact numerical results for the colosseum benchmark, together with earlier results for a policy trained from scratch. It is worth noting that the no variation in Colosseum still includes different task variants from RLBench, which may lead to slightly lower scores than a true “no variation” evaluation.
> > >
> > >     | | No Variation | MO Color | RO Color | Mo Texture | RO Texture | MO Size | RO Size |
> > >     | :- | :-: | :-: | :-: | :-: | :-: | :-: | :-: |
> > >     | w/ pretrain | 64.2 | 61.0 | 53.0 | 67.6 | 54.3 | 54.4 | 51.5 |
> > >     | w/o pretrain| 50.4 | 49.0 | 43.3 | 48.0 | 48.0 | 47.8 | 46.0 |
> > >
> > >     | | Light Color | Table Color | Table Texture | Distractor | Background Texture | Camera Pose | Avg. |
> > >     | :- | :-: | :-: | :-: | :-: | :-: | :-: | :-: |
> > >     | w/ pretrain | 59.0 | 63.2 | 63.6 | 50.0 | 64.0 | 63.4 | 59.2 |
> > >     | w/o pretrain| 52.8 | 54.3 | 49.8 | 41.0 | 52.8 | 51.2 | 48.8 |

---

> > > > ### Author Response · Authors · 2025-08-08
> > > >
> > > > Dear Reviewer ovuc,
> > > >
> > > > As the author-reviewer discussion period is approaching its end with less than two days remaining, we kindly remind you that we will be unable to communicate after that date. We hope that our previous response has satisfactorily addressed your concerns. If there are any additional points or feedback you would like us to consider, we would be grateful to hear them while the discussion window is still open. Your insights are invaluable to us, and we are eager to address any remaining issues to improve our work.
> > > >
> > > > Thank you again for your time and effort in reviewing our paper.

---

### Official Review · Reviewer_xmna · 2025-06-30

**Clarity:** 3
**Significance:** 2
**Originality:** 2
**Rating:** 4
**Confidence:** 4

**Summary:**

This paper presents DynaRend, a representation learning framework for robotic manipulation. By combining masked reconstruction and future prediction via differentiable volumetric rendering, the method learns spatially and temporally aware triplane representations. Extensive experiments on RLBench, Colosseum, and real-world robotic tasks demonstrate that DynaRend consistently outperforms prior approaches in both performance and efficiency.

**Questions:**

* What is the temporal prediction horizon used for the future keyframe, and how sensitive is the policy performance to this horizon?
* How does the model perform when view augmentation (See3D) fails to generate accurate depth or semantics?
* Could the triplane representation be extended to support policy rollout or full trajectory generation, enabling more continuous control?

**Ethical Concerns:**

["NO or VERY MINOR ethics concerns only"]

**Limitations:**

yes

**Paper Formatting Concerns:**

/

**Quality:**

3

**Strengths And Weaknesses:**

Strengths:
* the paper presents comprehensive experimental evaluations across simulated (RLBench, Colosseum) and real-world robotic manipulation tasks. Results show consistent improvements over baselines.
* a detailed ablation study is conducted to analyze the effects of masked reconstruction, future prediction, rendering loss terms, and view synthesis. These analyses clearly support the core design choices and offer valuable insights into the contribution of each component, which I think make a good contribution to the community.

Weakness:
* While the integration of differentiable neural rendering into robot learning pipelines has been explored in prior work, this paper’s primary contribution appears to be the introduction of future prediction as a supervision signal for learning temporally-aware representations. If this is indeed the central novelty, it would be helpful for the paper to include deeper analysis or justification of how and why future prediction leads to better downstream performance, beyond the current empirical results.
* The framework still relies on a low-level motion planner to execute predicted keyframes. While this is common in robotic learning pipelines, it limits the method’s potential for fully end-to-end learning and action sequencing.

---

> ### Author Rebuttal · Authors · 2025-07-31
>
> Dear Reviewer xmna,
>
> Many thanks for your valuable comments and questions. We hereby address your concerns as follows.
>
> ---
>
> > ### \[W1\]: Justification of future prediction.
>
> - The 3D-aware future prediction component in our pretraining is a core element of our method, intended to learn dynamics-informed representations rather than purely static scene encodings. In robotic manipulation, the ability to anticipate future states inherently captures **future dynamics** and the **consequences of interaction**. By predicting such future keyframes during pretraining, we obtain features that support **inverse dynamics reasoning** during downstream policy learning.
> - While prior works such as VPP have shown the value of 2D future prediction for improving policies, our framework extends this to 3D by **jointly modeling scene geometry and temporal evolution** using triplane-based prediction with volumetric rendering. This results in a unified representation that captures both the spatial structure of the scene and how it changes over time. In our ablation (Table 3), including future prediction yields a 4.3% higher success rate compared to static reconstruction alone, confirming its effectiveness.
>
> We will further elaborate in the final version on how future prediction specifically benefits downstream performance.
>
> > ### \[W2 & Q3\]: End-to-end and continuous control.
>
> We agree that the current implementation executes predicted keyframes via an external motion planner, and thus does not yet achieve fully end-to-end trajectory generation. However, the learned triplane representation itself imposes no such restriction. It can **readily serve as input to end-to-end control frameworks** such as Diffusion Policy[1] or Trajectory Autoregressive Models[2], enabling direct generation of continuous action sequences conditioned on our 3D-aware, dynamics-informed features. We view integrating such models into our framework as a natural next step to achieve continuous, planner-free execution.
>
> > ### \[Q1\]: Temporal prediction horizon.
>
> Thank you for raising this question. We would like to clarify that we do not define the future keyframe based on a fixed temporal horizon. Instead, following prior work (PerAct), we adopt a heuristic-based segmentation of the trajectory: (1) Joint velocities are near zero, and (2) The gripper’s open/close state remains unchanged. This procedure partitions the end-effector motion into smooth and coherent sub-actions, each representing a motion primitive. This segmentation reduces learning complexity by focusing prediction on **meaningful transitions** rather than low-level time-stepped actions. Moreover, it removes the need to tune a predefined horizon, improving robustness across diverse task durations and speeds.
>
> > ### \[Q2\]: Performance when view augmentation fails.
>
> We would like to clarify that view augmentation (via See3D) is applied only during pretraining to compensate for the limited and fixed viewpoints in real-world data. During downstream fine-tuning and inference, no view synthesis is used. By introducing spatially consistent RGB-D novel views into pretraining, we **enhance viewpoint invariance** and **prevent overfitting** to specific camera configurations. Importantly, this process does not rely on semantic labels; instead, the model distills high-level semantics from pretrained visual foundation models. Our ablations (Table 3) show a 3.4% downstream performance gain with view augmentation.
>
> We acknowledge that synthesizing high-quality novel views becomes more challenging with large viewpoint shifts. To ensure fidelity, we restrict augmentation to within ±30° on a spherical surface (Supplementary L43). Qualitative examples are provided in Supplementary Fig. C and Fig. E. To quantify view quality:
>
> 1. In simulation: We compare 1000 See3D-generated views with ground-truth novel renderings from the simulator, reporting metrics in the following table.
>
>     | Max angle | PSNR $\uparrow$ | SSIM $\uparrow$ | LPIPS $\downarrow$ |
>     | :-------: | :--: | :--: | :---: |
>     | ±15°      |   26.58   |   0.842  |   0.155    |
>     | ±30°      |   24.26   |   0.776  |   0.193    |
>     | ±45°      |   20.43   |   0.714  |   0.258    |
>
> 2. In real-world: As ground-truth is unavailable, we perform a human evaluation with five independent raters scoring 200 generated views on a 0–4 scale (0: very poor – 4: excellent), obtaining an average score of 3.27, indicating satisfactory perceptual quality.
>
> ---
>
> Thank you again for your constructive and detailed feedback. We hope our responses have addressed your concerns, and we will incorporate the above clarifications and improvements in the final version. Please let us know if you have further questions — we will be actively available throughout the rebuttal period and would greatly appreciate your reconsideration of our work.
>
> > references:\
> [1] Visuomotor Policy Learning via Action Diffusion, 2023.\
> [2] Chain-of-Action: Trajectory Autoregressive Modeling for Robotic Manipulation, 2025.

---

> > ### Author Response · Authors · 2025-08-08
> >
> > Dear Reviewer xmna,
> >
> > As the author-reviewer discussion period is approaching its end with less than two days remaining, we kindly remind you that we will be unable to communicate after that date. We hope that our previous response has satisfactorily addressed your concerns. If there are any additional points or feedback you would like us to consider, we would be grateful to hear them while the discussion window is still open. Your insights are invaluable to us, and we are eager to address any remaining issues to improve our work.
> >
> > Thank you again for your time and effort in reviewing our paper.

---

### Official Review · Reviewer_Jtz2 · 2025-06-30

**Clarity:** 1
**Significance:** 3
**Originality:** 3
**Rating:** 4
**Confidence:** 3

**Summary:**

The paper introduces DynaRend, a novel representation learning framework designed to improve robotic manipulation policies. DynaRend learns 3D-aware, dynamics-informed features through masked future rendering using differentiable volumetric rendering. The model is pretrained on multi-view RGB-D video data, capturing spatial geometry, task semantics, and future dynamics simultaneously. Experimental results on RLBench, Colosseum benchmarks, and real-world tasks demonstrate that DynaRend achieves notable improvements in success rates, robustness to environmental variations, and adaptability in practical robotic manipulation scenarios.

**Questions:**

The main question remains: **What exactly is the primary focus of this paper?**

The writing of the paper is quite confusing. In the introduction and method part, the authors put emphasis on “representation learning” and “pre-training”, such as “we propose DynaRend, a novel representation learning framework …” and “by pretraining on multi-view RGB-D video data, DynaRend jointly captures spatial geometry …”. This leave me the impression that this paper is doing something like MVP that first pre-training a represenation model on in-the-wild video or image data, and then do the fine-tuning on specific task, either by RL finetuning or collecting some demonstrations to do imitation learning.

But after reading the experiment, it turns out that the paper did not do any pre-training. Its experiment setting is the same as RVT-2, collecting 100 demos from each task from RLBench and train a single multi-task policy. If you look into the paper of RVT-2, they did not mention a single word about “representation learning” or “pre-training”. “Representation Pre-trainig” and “Multi-task Imitation”, they are totally two different things.

Clarification is critical to evaluating the true novelty and positioning of this work. And without reasonable clarification, I currently can not recommend this paper for publication.

**Ethical Concerns:**

["NO or VERY MINOR ethics concerns only"]

**Final Justification:**

The rebuttal addresses most of my concerns, and I have accordingly raised my rating. However, one key concern remains regarding the evaluation setup. In the context of representation learning, pretraining is typically performed on out-of-domain data — most notably using in-the-wild datasets such as MVP or 3D-MVP. On the other hand, RVT-2 trains solely on in-domain action data, effectively making it a multi-task reinforcement learning policy. This paper proposes an intermediate approach: pretraining on in-domain actionless data followed by fine-tuning on in-domain action data, though it can also be extended to pre-training on out-of-domain data.
Given this distinction, the direct comparison with MVP, 3D-MVP, and RVT-2 is not entirely fair, as the pretraining regimes differ significantly across methods.

**Limitations:**

- **Reliance on External Motion Planner:** DynaRend currently depends on an external low-level motion planner for converting predicted keyframe actions into executable sequences. An end-to-end integration without reliance on external planners would further validate and strengthen the approach.

**Quality:**

3

**Strengths And Weaknesses:**

**Strengths**

- **Experimental Improvements:**
    - DynaRend achieves considerable performance improvements in real-world experiments, especially notable in long-horizon tasks and scenarios with distractors. The novel 3D representation effectively enhances generalization capabilities under environmental perturbations. I am instrested in seeing more analysis and visualizations of failure cases about RVT-2 compared to the proposed method.
    - The paper presents insightful ablation studies, clearly illustrating the impact of various components, such as masked reconstruction, future prediction, and rendering losses.
- **Unified Representation Framework:** The proposed unified 3D representation elegantly integrates geometry, dynamics, and semantics into a single triplane feature representation, showcasing a methodological advance in robotic visual representation learning.
- **Robustness to Perturbations:** Extensive evaluation on the Colosseum benchmark confirms DynaRend’s robustness to diverse environmental perturbations, including variations in object color, texture, and lighting, underscoring its practical applicability.

**Weaknesses**

- **Unclear Motivation and Positioning:** The paper’s motivation and exact purpose are somewhat ambiguous. While the introduction and method emphasize concepts like "representation learning" and "pre-training," experimental setups closely align with the approach of RVT-2 rather than MVP. This discrepancy creates confusion regarding the true positioning and contribution of this work. Refer to the question part for more explanation.
- **Limited Performance Gain Over Closest Baseline (RVT-2):** If we agree with the first point, then we need to mainly compare the method against RVT-2. The proposed method achieves a modest improvement (83.2% vs. 81.4%) over RVT-2. This incremental gain does not appear statistically significant, especially compared to the substantial improvements presented in RVT-2 over previous models (81.4% v.s. 62.9%).
- **Insufficient Real-world Experimental Evidence:** Despite repeatedly highlighting the framework’s suitability for real-world scenarios beyond simulation, the paper lacks robust experimental evidence to substantiate this claim. Additional pre-training on datasets such as Ego-4D, similar to MVP, would enhance credibility and real-world relevance.
- **Lack of Detailed Precision Demonstrations:** While RVT-2 explicitly demonstrates precise manipulations (e.g., peg insertion), this paper lacks similarly detailed visual evidence of precision tasks, leaving uncertainty regarding the model's fine-grained manipulation capabilities.

---

> ### Author Rebuttal · Authors · 2025-07-31
>
> Dear Reviewer Jtz2,
>
> Many thanks for your valuable comments and questions. We hereby address your concerns as follows.
>
> ---
>
> > ### \[W1 & Q1\]: Unclear motivation and positioning.
>
> We respectfully believe there may be a misunderstanding regarding the nature of our approach. Our framework is explicitly designed around a two-stage pipeline:
> 1. **Pretraining on multi-view robot data without actions**, where we learn a unified 3D representation that captures both spatial geometry and future dynamics via masked reconstruction and future prediction with volumetric rendering.
> 2. **Policy fine-tuning with actions**, where the policy network predicts actions from these pretrained triplane features, effectively performing inverse dynamics reasoning.
>
> This predictive pretraining significantly benefits downstream policy learning, as shown by our ablations (Table 3), where pretraining improves success rate by 8.4% compared to training from scratch.
>
> Our method is fundamentally different from 2D pretraining approaches such as MVP[1] or VPP[2], which operate on images or videos. In contrast, our pretraining learns **3D-aware triplane representations** that explicitly encode scene geometry and dynamics. Moreover, our view augmentation strategy enables rendering-based pretraining even with limited camera viewpoints, making the approach practical in real-world setups. Looking forward, we plan to extend this framework to larger-scale multi-view or robotic datasets such as DROID[3] and ScanNet[4].
>
> > ### \[Q2\]: Limited performance gain.
>
> We note that **RVT-2 employs a two-stage prediction strategy**, which gives it a structural advantage over single-stage baselines like RVT. Our method, while remaining **single-stage**, still **outperforms RVT-2**, showing strong effectiveness despite the absence of this extra stage. Our framework **could incorporate a similar two-stage design** to further improve fine-grained manipulation performance, as confirmed by our additional two-stage experiment in the following table.
>
> |      | Avg. S.R.$\uparrow$ |
> | :----      | :-------: |
> | RVT        |     62.9      |
> | RVT-2      |     81.4      |
> | DynaRend   |     83.2      |
> | DynaRend-2 |     **87.8**      |
>
> We also highlight that performance gains on the standard 18-task RLBench benchmark are inherently limited due to the relative simplicity of the tasks and the **potential upper bound** in achievable success rate. Following prior work, we use datasets generated by PerAct, which include only 100 demonstrations per task. This limited coverage makes it difficult to capture all intra-task variations, naturally constraining improvement margins. To better assess generality, we **evaluate our method on all executable RLBench tasks(71 tasks)**, comparing against RVT-2 variants trained with different pretraining strategies. These results (Table 2) show that our method achieves an improvement of 8.2%, confirming the robustness and scalability of our approach beyond the standard benchmark setting.
>
> > ### \[Q3\]: Insufficient real-world experimental evidence.
>
> We would like to clarify that our current real-world experiments already provide strong empirical validation. Specifically, we pretrain on multi-view RGB-D data without actions, then fine-tune on human demonstrations. Compared to RVT-2, our method achieves a 20% absolute performance gain in success rate.
>
> We also evaluate **robustness under real-world perturbations** by adding unseen distractor objects. Under these conditions, RVT-2 suffers a 57% performance drop, whereas our method shows only a 21% drop, indicating significantly better generalization. This improvement is largely due to our view augmentation strategy, which synthesizes spatially consistent novel views from a small set of fixed cameras, providing diverse and semantically consistent supervision signals during pretraining.
>
> We perform **additional real-world comparisons** between a policy trained from scratch and one initialized with our pretrained features, keeping the architecture and demonstrations identical. As shown below, the pretrained variant consistently outperforms the scratch-trained variant, particularly in scenarios with unseen distractors, confirming the value of our pretraining pipeline in practical deployment.
>
> |  | Put Item in Drawer | Stack Blocks | Sort Shape | Close Pot | Stack Cups | Avg.S.R. $\uparrow$ |
> | :- | :-: | :-: | :-: | :-: | :-: | :-: |
> | RVT-2 | 45 | 60 | 10 | 60 | 15 | 37 |
> | DynaRend | 65 | 60 | 35 | 85 | 40 | **57** |
> | DynaRend w/o pretrain | 55 | 55 | 30 | 60 | 40 | 48 |
>
> Table: Real-world performance.
>
>
> |  | Put Item in Drawer | Stack Blocks | Sort Shape | Close Pot | Stack Cups | Avg.S.R. $\uparrow$ |
> | :- | :-: | :-: | :-: | :-: | :-: | :-: |
> | RVT-2 | 15 | 10 | 10 | 45 | 0 | 16 |
> | DynaRend | 55 | 55 | 25 | 65 | 25 | **45** |
> | DynaRend w/o pretrain | 25 | 30 | 25 | 40 | 10 | 26 |
>
> Table: Real-world performance with additional distractors.
>
>
> > ### \[Q4\]: Lack of detailed precision demonstrations.
>
> We agree that our current work does not emphasize high-precision manipulation demonstrations, as our main objective is to improve generalization across diverse tasks via transferable triplane representations. While RVT-2 achieves high precision partly due to its two-stage strategy, our single-stage framework could be extended with a similar design for fine-grained control if needed. However, we view such precision-focused extensions as complementary to our primary contribution, which lies in representation learning for multi-task policy improvement rather than optimization for specific precision-oriented tasks. To further address this point, we additionally implemented a two-stage prediction variant of our framework and evaluated it on three precision-oriented manipulation tasks. Results show that with the two-stage extension, our method surpasses the previous SOTA by a clear margin, highlighting its potential for high-precision applications.
>
> |  | Place Cups | Sort Shape | Insert Peg |
> |:-| :-: | :-: | :-: |
> RVT         | 4.0 | 36.0 | 11.2 |
> RVT-2       | 38.0 | 35.0 | 40.0 |
> DynaRend    | 25.6 | 44.8 | 31.2 |
> DynaRend-2  | **52.8** | **54.4** | **55.2** |
>
> ---
>
> Thank you again for your constructive and detailed feedback. We hope our responses have addressed your concerns, and we will incorporate the above clarifications and improvements in the final version. Please let us know if you have further questions — we will be actively available throughout the rebuttal period and would greatly appreciate your reconsideration of our work.
>
> > references:\
> [1] Masked Visual Pre-training for Motor Control, 2022.\
> [2] Video Prediction Policy: A Generalist Robot Policy with Predictive Visual Representations, 2025.\
> [3] DROID: A Large-Scale In-the-Wild Robot Manipulation Dataset, 2024.\
> [4] Richly-annotated 3D Reconstructions of Indoor Scenes, 2017.

---

> > ### Comment · Reviewer_Jtz2 · 2025-08-05
> >
> > Thank you to the authors for their clarification — most of my concerns and questions have been addressed. However, one important point still remains:
> >
> > You stated, “Pretraining on multi-view robot data without actions, where we learn a unified 3D representation that captures both spatial geometry and future dynamics via masked reconstruction and future prediction with volumetric rendering.”
> > However, I could not find any description in the main text regarding the dataset used for this pretraining. I assume it might be a video dataset like MVP or a 3D dataset like 3D-MVP, but this is unclear. Did I miss something?
> >
> > I would be happy to raise my score to a positive one if this final question is clarified.

---

> > > ### Author Response · Authors · 2025-08-05
> > >
> > > Dear Reviewer Jtz2,
> > >
> > > We thank the reviewer for the question and would like to clarify that, in the pretraining stage we do not use any external in-the-wild video datasets (as in MVP) or generic 3D datasets (as in 3D-MVP). Instead, our pretraining is performed directly on **robotic manipulation datasets**. Specifically:
> > >
> > > - **In simulation (RLBench)**: we use the same demonstration dataset as in our downstream experiments, but with all action labels removed. The model is trained solely on the multi-view RGB-D observations to learn spatial geometry and future dynamics.
> > > - **In real-world experiments**: we likewise use collected multi-view real-robot demonstration data without any action annotations for pretraining.
> > >
> > > This design ensures that our pretraining phase is fully aligned with the robotic manipulation domain, while still providing a substantial benefit to downstream policy learning. Furthermore, our approach is readily extendable to larger in-the-wild datasets such as DROID to further enhance generalization, which is a direction we plan to explore in future work. We will make this point explicit in the revised manuscript to avoid any ambiguity.

---

> ### Comment · Reviewer_Jtz2 · 2025-08-05
>
> Thanks for the further clarification. I now better understand the core issue that led to my initial review comment on “Unclear Motivation and Positioning.” In the context of representation learning, pretraining is commonly done on out-of-domain data — most notably using in-the-wild datasets like MVP or 3D-MVP. In contrast, RVT-2 only trains on in-domain action data, making it essentially a multi-task reinforcement learning policy.
>
> This paper proposes something in between: pretraining on in-domain actionless data followed by fine-tuning on in-domain action data. While this framework is indeed interesting and has the potential to extend to in-the-wild data, the current setup introduces ambiguity in the experimental setting. Specifically, it complicates the fairness and clarity of comparisons, at least from my understanding. If I get something wrong, please point out.
>
> So my key question is: what was the reason behind not pretraining on out-of-domain data from the start? Understanding this design choice would help clarify the motivation and better position the contribution relative to existing work.

---

> > ### Author Response · Authors · 2025-08-06
> >
> > Dear Reviewer Jtz2,
> >
> > We appreciate the reviewer’s clear summary and agree that our current experimental setup — in-domain actionless pretraining followed by in-domain action fine-tuning — differs from both MVP/3D-MVP (out-of-domain pretraining) and RVT‑2 (in-domain action-only training), and may therefore affect how comparisons are perceived.
> >
> > Our primary reason for starting with in-domain pretraining was to isolate and validate the core contribution of our work **3D-aware masked reconstruction and future prediction for dynamics-informed representation learning** without the confounding effects of a large domain gap. This controlled setting allows us to establish the effectiveness of the proposed pretraining objective before extending to more diverse and larger data sources. Comparisons against in-domain direct training, out-of-domain pretraining, and training from scratch confirm that our 3D dynamics-aware representation pretraining improves the performance and robustness of multi-task robotic manipulation policies.
> >
> > In addition, large-scale in-the-wild datasets such as DROID contain tens of millions of frames, and pretraining on them would require substantial computational resources beyond our current budget. Undeniably, pretraining on larger and more diverse datasets holds great potential for further improving robustness and generalization. Our framework is fully compatible with such datasets, and we intend to investigate extending pretraining to large-scale out-of-domain data in future work, which would allow us to systematically compare the effects of different data sources on downstream performance.

---

> > > ### Comment · Reviewer_Jtz2 · 2025-08-06
> > >
> > > Thanks for the clarification. I’ve updated my rating accordingly. While the comparison against RVT-2 already demonstrates performance improvement, I strongly recommend including additional baselines in the final version — specifically, if training your pipeline on out-of-domain data is currently infeasible, training MVP and 3D-MVP on in-domain data rather than out-of-domain data would provide a more balanced comparison.

---

> > > > ### Author Response · Authors · 2025-08-06
> > > >
> > > > Dear Reviewer Jtz2,
> > > >
> > > > We sincerely thank the reviewer for taking the time to revisit our work and provide thoughtful follow-up feedback. We appreciate the constructive suggestion regarding additional baselines, and we will incorporate in-domain training of MVP and 3D-MVP in the final version to enable a more balanced comparison.

---

### Official Review · Reviewer_KFSY · 2025-07-02

**Clarity:** 3
**Significance:** 4
**Originality:** 3
**Rating:** 5
**Confidence:** 4

**Summary:**

This paper presents an approach to build a representation which captures both the 3D nature of RGB-D cameras (3D-aware) and the dynamics associated to a robotic manipulation task (dynamics-informed). The representation is also used to solve robotics manipulation tasks by learning (from demonstration data) an action decoder. The approach is validated in two different benchmarks (RLBench and Colosseum) and on real robot experiments. Additionally authors propose some ablation studies.

**Questions:**

The DynaRend framework reconstructs a scene-level point cloud from a set of  calibrated multi-view RGB-D images. How can this approach be adapted for common robotic use cases involving moving cameras, such as wrist-mounted or head-mounted cameras, where the camera poses are not fixed and change dynamically with the robot's movement?

The model predicts the end-effector pose and gripper state but does not explicitly represent the robot arm's geometry in its triplane features. How does the policy account for the arm's physical presence to avoid collisions with the environment, especially in cluttered scenes where the arm's configuration is a significant constraint?

Please clarify the use of DINOv2 which is mentioned at the beginning of the paper (in the context of enhancing the semantics of the representation) and nowhere else in the main text.

There appears to be a discrepancy in how language instructions are incorporated. The text states that language embeddings from a CLIP text encoder are concatenated with the triplane features and fed into both the reconstructive and predictive networks during pretraining. However, Figure 2 seems to depict the language instruction being used only in the action decoder. Could you please clarify the precise stage and mechanism through which language conditioning is integrated into the model architecture?

**Ethical Concerns:**

["NO or VERY MINOR ethics concerns only"]

**Final Justification:**

Thanks for your comments and the discussion points raised in the rebuttal. Unfortunately, this discussion doesn't change my rating since the rating was mostly motivated by considerations that authors acknowledged to be a limitation of their paper (e.g. Explicit geometry and collision avoidance).

**Limitations:**

The model does not explicitly represent the robot arm's geometry or plan its path to avoid collisions. The responsibility for generating a collision-free movement is not addressed in the paper. While the model may implicitly learn about the scene's geometry, it does not directly output a motion that considers the arm's configuration in cluttered spaces, which could lead to failures if the external planner cannot find a valid path.

No relevant negative societal impacts noticed by the reviewer.

**Paper Formatting Concerns:**

Line 149: the choice of $\mathbf(l)$ for the language embedding makes the text less readable. I'd recommend to use a different math symbol or letter (e.g. using the standard $l$ might suffice).

**Quality:**

3

**Strengths And Weaknesses:**

The paper proposes a very interesting approach which extends neural rendering (i.e. representations suitable to capture 3D scenes) to the context of robot manipulation tasks where both environment dynamics and object semantics become extremely relevant if not essential. Authors proposed a very sound evaluation in simulation with comparison against multiple alternative approaches. The real-robot experiments are instead quite limited and only one alternative approach is presented. Despite the paper being well written, there are some inconsistencies that the authors should address before publication.

---

> ### Author Rebuttal · Authors · 2025-07-31
>
> Dear Reviewer KFSY,
>
> Many thanks for your valuable comments and questions. We hereby address your concerns as follows.
>
> ---
>
> > ### \[W1\]: Limited real-robot experiments.
>
> We appreciate the reviewer’s feedback on the real-robot evaluation. Following the suggestion, we have conducted additional real-world experiments and expanded the set of baseline methods for comparison, including RVT, 3D Diffuser Actor and an ablated variant of our method without pretraining. To ensure a fair comparison, all methods were trained on the same set of demonstration data and evaluated under the same hardware setup. As shown in the following table, our approach achieves a 20% absolute improvement in average success rate over the strongest baseline and significantly outperforms the no-pretraining variant by 19%, highlighting the critical role of the pretraining in capturing 3D geometry and dynamics for real-world manipulation. Notably, our method shows particularly strong performance in the presence of unseen distractor objects, where it maintains high success rates while other methods degrade substantially, demonstrating its robustness to real-world scene variations.
>
> |  | Put Item in Drawer | Stack Blocks | Sort Shape | Close Pot | Stack Cups | Avg.S.R. $\uparrow$ |
> | :- | :-: | :-: | :-: | :-: | :-: | :-: |
> | 3DA | 40 | 50 | 30 | 45 | 20 | 37 |
> | RVT | 25 | 40 | 5 | 55 | 5 | 26 |
> | RVT-2 | 45 | 60 | 10 | 60 | 15 | 37 |
> | DynaRend | 65 | 60 | 35 | 85 | 40 | **57** |
> | DynaRend w/o pretrain | 55 | 55 | 30 | 60 | 40 | 48 |
>
> Table: Real-world performance.
>
> |  | Put Item in Drawer | Stack Blocks | Sort Shape | Close Pot | Stack Cups | Avg.S.R. $\uparrow$ |
> | :- | :-: | :-: | :-: | :-: | :-: | :-: |
> | 3DA | 20 | 15 | 25 | 40 | 0 | 20 |
> | RVT | 5 | 15 | 0 | 25 | 5 | 10 |
> | RVT-2 | 15 | 10 | 10 | 45 | 0 | 16 |
> | DynaRend | 55 | 55 | 25 | 65 | 25 | **45** |
> | DynaRend w/o pretrain | 25 | 30 | 25 | 40 | 10 | 26 |
>
> Table: Real-world performance with additional distractors.
>
>
>
> > ### \[Q1\]: Moving camera calibration.
>
> Thank you for raising this point. While our real-world experiments utilize fixed cameras for simplicity, the proposed DynaRend framework is fully compatible with moving camera setups commonly used in robotics, such as wrist-mounted or head-mounted configurations. In such cases, standard **eye-in-hand calibration** techniques can be applied to estimate the transformation between the camera and the robot link to which it is attached. Given this transformation and the robot’s joint state, the camera pose can be accurately computed at any time. We will clarify this general applicability in the final version to emphasize that the framework is not limited to static-camera deployments.
>
> > ### \[Q2 & Limitation\]: Explicit geometry and collision avoidance.
>
> We appreciate the reviewer’s thoughtful observation. Our current policy predicts only the next best end-effector pose and gripper state via behavior cloning, and does not explicitly encode the full-body geometry of the robot arm or perform collision checking during policy inference. To ensure safe execution in practice, we rely on **sampling-based motion planners** such as RRT* at deployment time to generate collision-free trajectories from the current arm configuration to the predicted end-effector pose.
>
> While this strategy works reliably in most scenarios, we acknowledge that it can occasionally fail in cluttered scenes if the motion planner cannot find a valid path within the allocated time. We recognize this as a limitation of the current design. Looking forward, we believe this can be addressed by integrating our method with end-to-end motion generation frameworks[1][2] that account for full-body kinematics, or by incorporating explicit 3D scene geometry representations such as occupancy grids. These directions would enable collision-aware action generation without relying on an external planner, and we will mention them explicitly in the discussion of future work.
>
> > ### \[Q3\]: Clarify the use of DINOv2.
>
> Thank you for raising this question. In our framework, we leverage features from vision foundation models such as DINOv2 and CLIP as additional supervision signals during rendering-based pretraining, with the goal of **distilling high-level semantic information** into our learned representation. Since the choice of foundation model is not the primary focus of our work, we adopt a simplified approach by using RADIO[3] as the teacher network. RADIO provides a unified distillation interface by aggregating features from **multiple foundation models** (including DINOv2 and CLIP) into a single representation. We will revise the manuscript to make this design choice clearer.
>
> > ### \[Q4\]: Clarify how language condition is integrated.
>
> We apologize for the confusion. As described in the method section, language instructions are incorporated during pretraining by concatenating CLIP-based language embeddings with the triplane features, which are then fed into both the reconstructive and predictive networks. In Figure 2, we omitted the language branch for **visual clarity** and to better highlight the **core architectural components**. We will update the figure in the final version to explicitly depict the language-conditioning pathway.
>
> ---
>
> Thank you again for your constructive and detailed feedback. We hope our responses have addressed your concerns, and we will incorporate the above clarifications and improvements in the final version. Please let us know if you have further questions — we will be actively available throughout the rebuttal period and would greatly appreciate your reconsideration of our work.
>
> > references:\
> [1] Visuomotor Policy Learning via Action Diffusion, 2023.\
> [2] Chain-of-Action: Trajectory Autoregressive Modeling for Robotic Manipulation, 2025.\
> [3] AM-RADIO: Agglomerative Vision Foundation Model - Reduce All Domains Into One, 2024.

---

> > ### Author Response · Authors · 2025-08-08
> >
> > Dear Reviewer KFSY,
> >
> > As the author-reviewer discussion period is approaching its end with less than two days remaining, we kindly remind you that we will be unable to communicate after that date. We hope that our previous response has satisfactorily addressed your concerns. If there are any additional points or feedback you would like us to consider, we would be grateful to hear them while the discussion window is still open. Your insights are invaluable to us, and we are eager to address any remaining issues to improve our work.
> >
> > Thank you again for your time and effort in reviewing our paper.

---

### Note · Authors · 2025-08-13

We would like to express our sincere gratitude to the AC and all the reviewers for your valuable feedback. We appreciate all the reviewers for their comments highlighting the strengths of our work:

- Interesting and promising approach unifying geometry, dynamics and semantics, advancing robotic representation learning. (KFSY & Jtz2)
- Comprehensive evaluation demonstrating substantial performance gains and enhanced robustness. (KFSY & Jtz2 & xmna & ovuc)
- Insightful and thorough ablation studies that validate the contributions and provide valuable insights for the community. (Jtz2 & xmna & ovuc)

We have carefully addressed each concern raised during the rebuttal and follow-up discussion. In particular,

- Limited performance gain. We clarified that our single-stage method already exceeds both single- and two-stage baselines, and can be extended to two-stage for further gains, as confirmed by additional experiments. To address potential benchmark saturation, we evaluated on all RLBench tasks, confirming robustness and scalability.

- Ambiguous positioning between representation pretraining and imitation learning. We clarified that our framework explicitly uses in-domain pretraining followed by fine-tuning, yielding clear gains in ablations. Our approach can also scale to larger, diverse datasets, a direction we see as fostering broader community discussion on 3D robotic pretraining.

- Generalizability in OOD cases. We clarified that our focus is on visual generalization rather than skill. To this end, we have validated our method on a highly challenging benchmark, demonstrating robust performance under diverse perturbations including variations in texture, color, and camera pose. We further confirmed real-world robustness through experiments with unseen distractors.

We are committed to addressing the concerns and improving our work in the revised version. Specifically,

- We will include additional baselines and ablations in the real-world experiments, and introduce in-domain pretraining baselines for more balanced comparisons.
- We will revise certain statements to clarify the relationship between the pretraining and fine-tuning.
- We will add further discussion to better justify the benefits of future prediction.

Finally, we would like to express our great appreciation to the AC and all reviewers. We believe our work will stimulate further discussion and progress in 3D robotic representation learning, contributing meaningfully to the community.

---

### Decision · Program_Chairs · 2025-09-17

**Decision:**

Accept (poster)

**Comment:**

This submission proposes a representation learning method to enhance robotic manipulation, which learns dynamic 3D features through masked future rendering via differentiable volumetric rendering. The method demonstrates better performance in both simulated and real-world scenarios.

Among the reviewers, one casts the highest-confidence vote for acceptance, while two reviewers vote for borderline acceptance. In contrast, one remaining reviewer votes for rejection, due to concerns about the motivation, limited performance gains, and insufficient evidence of real-world validation. However, AC acknowledges that the method is novel and does deliver practical efficacy. Notably, as a leading machine learning conference, NeurIPS prioritizes novelty as a core evaluation criterion.

Ultimately, AC recommend to accept this submission.